# Landmark-RxR: Solving Vision-and-Language Navigation with Fine-Grained Alignment Supervision

Keji He[1,2]    Yan Huang[1,2]    Qi Wu[3]    Jianhua Yang[5]    Dong An[1,4]    Shuanglin Sima[1,2]
Liang Wang[*1,2,6,7]

[1]Center for Research on Intelligent Perception and Computing
National Laboratory of Pattern Recognition
Institute of Automation, Chinese Academy of Sciences
[2]School of Artificial Intelligence, University of Chinese Academy of Sciences
[3]School of Computer Science, University of Adelaide
[4]School of Future Technology, University of Chinese Academy of Sciences
[5]School of Artificial Intelligence, Beijing University of Posts and Telecommunications
[6]Center for Excellence in Brain Science and Intelligence Technology (CEBSIT)
[7]Chinese Academy of Sciences, Artificial Intelligence Research (CAS-AIR)
{keji.he, dong.an, shuanglin.sima}@cripac.ia.ac.cn    {yhuang,
wangliang}@nlpr.ia.ac.cn    qi.wu01@adelaide.edu.au    youngjianhua@bupt.edu.cn

## Abstract

In Vision-and-Language Navigation (VLN) task, an agent is asked to navigate inside 3D indoor environments following given instructions. Cross-modal alignment is one of the most critical challenges in VLN because the predicted trajectory needs to match the given instruction accurately. In this paper, we address the cross-modal alignment challenge from the perspective of fine-grain. Firstly, to alleviate weak cross-modal alignment supervision from coarse-grained data, we introduce a human-annotated fine-grained VLN dataset, namely Landmark-RxR. Secondly, to further enhance local cross-modal alignment under fine-grained supervision, we investigate the focal-oriented rewards with soft and hard forms, by focusing on the critical points sampled from fine-grained Landmark-RxR. Moreover, to fully evaluate the navigation process, we also propose a re-initialization mechanism that makes metrics insensitive to difficult points, which can cause the agent to deviate from the correct trajectories. Experimental results show that our agent has superior navigation performance on Landmark-RxR, en-RxR and R2R. Our dataset and code are available at https://github.com/hekj/Landmark-RxR.

## 1 Introduction

Vision-and-language navigation (VLN) is an important task about cross-modal intelligence and can be applied to service and rescue robots. Different from other cross-modal tasks like image/video captioning [1, 2] and visual question answering [3], where the agent only needs to understand fixed images or videos, VLN agent has to learn and reason by dynamically interacting with the real environment guided by human instructions. Since Anderson *et al.* [4] first introduced the VLN task with the coarse-grained dataset Room-to-Room (R2R), great progresses in this direction have been made, ranging from the sequence-to-sequence models [4] combined with cross-modal grounding modules [5–8] to Transformer-based models [7, 9–12].

Cross-modal alignment is one of the most critical challenges in VLN as the predicted trajectory needs to match the given instruction accurately. However, it is difficult for the agent to learn very

---

*Corresponding author

35th Conference on Neural Information Processing Systems (NeurIPS 2021).

accurate visual and textual modality alignment from coarse-grained data, *i.e.*, only coarse-level correspondences between global instructions and trajectories are annotated, without finer ones between sub-instructions and sub-trajectories. Works mentioned above [4–8, 10–12] tried attending on specific parts of the instructions during navigation to help the agent better align salient visual contents with the textual modality. But only coarse-grained data like R2R are not capable of providing enough supervising signals of accurate cross-modal alignment. Some recent works such as Fine-Grained R2R [13] and BabyWalk [14] also found this same issue and proposed fine-grained datasets based on R2R. However, their datasets are generated by heuristic rules and are not precise enough, which inevitably limit the navigation performance. Although Room-across-Room (RxR) dataset [15] includes word-level alignment between each word and point in the trajectory, it lacks segmentation tags to split an instruction into sub-instructions with independent meanings and to figure out the landmarks in a trajectory, namely the marked end points of sub-trajectories. Since words in the same sub-instruction are highly correlated, these important correlations are lost because no sub-instruction exists in RxR. It is the same for the sub-trajectories. As a result, there is also an absence of the correlation between the words in a sub-instruction and the points in corresponding sub-trajectories, which should be an important cross-modal alignment supervision signal.

In addition to the dataset annotation, designing better rewards during reinforcement learning is another important issue on cross-modal alignment learning. Currently, reward shaping has been well investigated in VLN based on coarse-grained data to align instructions and trajectories globally, but pays little attention to enhance local cross-modal alignment. The goal-oriented reward simply [6] uses the arrival signal and the reduced distance to the goal point as the reward, guiding agent to find the global goal points. The fidelity-oriented rewards [16, 17] help agent improve the similarity between the predicted trajectory and demonstration trajectory globally by taking intermediate points as external supervision. However, some of the points in the trajectory have more detailed descriptions and are more helpful to navigation process, *i.e.*, they can assist agent in knowing the visual scenes around these specific points in the trajectories could be better aligned with certain parts of the instructions than the trivial points, where the alignment is in a local to local manner. These points are called critical points in this paper. Thus, an emphasis on these critical points in the reward shaping for better local cross-modal alignment is essential.

In this paper, we address the challenges above from the perspective of fine-grain. Firstly, based on the English Guide part of RxR (en-RxR), we introduce a fine-grained dataset Landmark-RxR, which is human-annotated, landmark-based, fine-grained and currently the largest scale. With the groundtruth fine-grained data, experiments demonstrate that agent generalizes better to unseen environments and instructions with domain gap. This indicates that the fine-grained data help agent to align textual and visual modalities better. Secondly, we propose two kinds of focal-oriented rewards that encourage local alignment between instructions and critical points. Since the landmarks in Landmark-RxR naturally meet the requirements of critical points, we just sampled critical points from the landmark set in Landmark-RxR. The focal-oriented rewards outperform the commonly used goal-oriented reward and fidelity-oriented reward. We also propose the re-initialization mechanism to fully evaluate the navigation process in a way that is insensitive to difficult points, which can cause the deviation from the correct trajectory. With the fine-grained data and focal-oriented rewards, our agent shows superior navigation performance on Landmark-RxR, en-RxR and R2R.

## 2   Related Work

**Vision-and-Language Navigation**   Anderson *et al.* [4] first introduced the simulated, photo-realistic VLN task with a benchmark Room-to-Room (R2R) and several sequence-to-sequence baselines. Fried *et al.* [5] utilized the speaker-driven data and panoramic action space to improve navigation performance. Wang *et al.* [18] proposed a planned-ahead hybrid reinforcement learning model with model-free and model-based reinforcement learning for better generalization. Tan *et al.* [19] first trained agent with mixed imitation learning and reinforcement learning and used unseen triplets generated by environment dropout method for fine-tuning then. Jain *et al.* [16] and Ilharco *et al.* [17] proposed the fidelity-oriented rewards to ensure the similarity between predicted trajectories and demonstration trajectories. Kurita *et al.* [20] combined generative and discriminative policy to enhance the navigation ability from different aspects. An *et al.* [21] fused neighbor visual contexts both globally and locally to get visual features with richer semantics. Huang *et al.* [22] defined two in-domain auxiliary tasks for representation learning which benefit the downstream navigation task. Li *et al.* [7] leveraged the large-scale pretrained language models BERT [23] and GPT [24]

for powerful text representation. Hao *et al.* [10] pretrained and finetuned generic transformer-based model in VLN environment and the pretrained model generalizes well to other two navigation tasks. Majumdar *et al.* [11] demonstrated that pretrained on image-text pairs from the web, the transformer-based model can perform on VLN task well after fine-tuning. Hong *et al.* [12] proposed a Recurrent Vision-and-Language BERT to process time-dependent tasks like VLN efficiently.

**Fine-Grained VLN**   Wang *et al.* [6] designed a cross-modal grounding module to infer which part of the instructions to focus on. Ma *et al.* [8] introduced a self-monitoring agent with a visual-textual co-grounding module to locate the required part in instructions and a progress monitor to reflect navigation progress. Qi *et al.* [25] decoupled instructions into object-aware and action-aware parts for more accurate action prediction. Zhu *et al.* [14] adopted a two-stage method for navigation. The given instruction is first split into several sub-instructions named baby-steps according to heuristic rules and then the baby-steps which are much easier to navigate are fed to the agent one by one during validation. Similar to [14], Hong *et al.* [13] first segmented instruction into sub-instructions using a heuristic method. Then the agent navigates following these easy sub-instructions sequentially with a shifting module inferring whether current sub-instruction has been completed. Ku *et al.* [15] explored utilizing spatially-temporally aligned annotations to supervise the textual and visual attention weights but got very few performance improvements. Different from these works, we propose a human-annotated, fine-grained dataset Landmark-RxR to alleviate the weak cross-modal alignment supervision from the coarse-grained data. Our experiments is designed to demonstrate the supervision from fine-grained and coarse-grained data can complement each other to improve the cross-modal alignment ability of the model itself. In addition, two kinds of focal-oriented rewards using fine-grained supervision signals are proposed to enhance the local cross-modal alignment by paying more attention to critical points which benefit the navigation process more.

## 3   Landmark-RxR Dataset

Our Landmark-RxR is built based on the English Guide split of RxR (en-RxR). It contains sub-instruction and sub-trajectory pairs (sub pairs) split from instructions in en-RxR. To facilitate the collection of sub pairs, we develop a 3D web-based collection tool. It allows the annotator to mark the landmarks, namely the end points of sub-trajectories, in trajectories and split out the corresponding sub-instructions. During annotation, the annotators can move between the discrete points or change heading and evaluation continuously by mouse click, like what the agent will do in the VLN task. In the following, we will describe the data collection and dataset analysis of Landmark-RxR.

### 3.1   Data Collection

The annotators are asked to read the complete instructions and then explore the 3D environments following corresponding trajectories. If landmarks are found, annotators need to mark them and split out the corresponding sub-instructions by interacting with the collection tool. As illustrated in Figure 1 (a), the complete instruction and complete trajectory pair (complete pair) are sampled from en-RxR. In Figure 1 (b), three landmarks in the complete trajectory are marked and the corresponding sub-instructions are also split out by annotators. These three sub pairs will be included in our Landmark-RxR dataset. Five principles are proposed to ensure the annotation process more standardized.

**Matching Principle**: The sub-trajectory should match the sub-instruction accurately.

**Independent Meaning Principle**: Each sub-instruction expresses an independent meaning.

**Position Change Principle**: The end point of each sub-trajectory should differ from the start point.

**Clear End Point Principle**: The end point should be described clearly in a sub-instruction. For example, the sub-instruction $go\ toward\ the\ door$ has a clear description of end point $door$, but no end point is described in the sub-instruction $go\ forward$.

**Minimum Granularity Principle**: The sub-instruction and sub-trajectory pair should be the minimum granularity which means it can not be split anymore.

After an annotator finishes the annotations, another annotator will verify the annotations again and modify the inaccurate part to ensure the annotations satisfy the five principles. For some mistakes in RxR, we also correct them in Landmark-RxR. Totally 30 annotators participated in the annotation task, contributing about 2,700 hours.

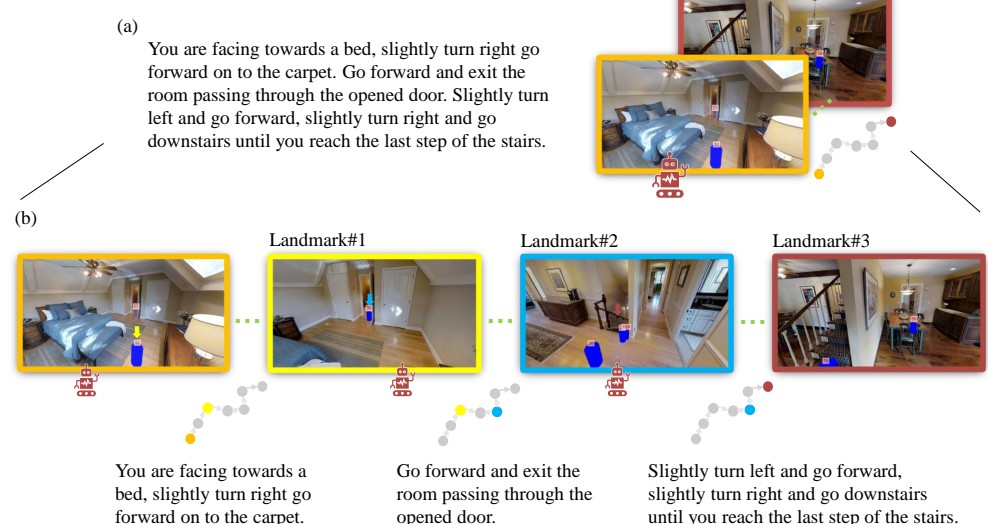

(a) You are facing towards a bed, slightly turn right go forward on to the carpet. Go forward and exit the room passing through the opened door. Slightly turn left and go forward, slightly turn right and go downstairs until you reach the last step of the stairs.

(b)

Landmark#1      Landmark#2      Landmark#3

You are facing towards a bed, slightly turn right go forward on to the carpet.

Go forward and exit the room passing through the opened door.

Slightly turn left and go forward, slightly turn right and go downstairs until you reach the last step of the stairs.

Figure 1: Three landmarks are marked in the complete trajectory and the corresponding sub-instructions are split out from the complete instruction. (a) An example of complete instruction and complete trajectory pair in en-RxR. (b) An example of three sub-instruction and sub-trajectory pairs split out from the complete instruction and complete trajectory pair. Each sub-instruction and sub-trajectory pair will be included in Landmark-RxR.

## 3.2 Dataset Analysis

Table 1: Statistics on R2R, RxR, en-RxR and our Landmark-RxR. Edge means the number of edges each trajectory contains on average.

| Dataset | Edge | Instruction | | | | | Trajectory |
| --- | --- | --- | --- | --- | --- | --- | --- |
| | | Train | Validation Seen | Validation Unseen | Test | Total | |
| R2R [4] | 5 | 14,025 | 1,020 | 2,349 | 4,173 | 21,567 | 7,189 |
| RxR [15] | 8 | 79,467 | 8,813 | 13,625 | 24,164[1] | 126,069 | 16,522 |
| en-RxR [15] | 8 | 26,464 | 2,939 | 4,551 | - | 33,954 | 11,321 |
| Landmark-RxR | 1.6 | 133,602 | 13,591 | 19,547 | - | 166,740 | 46,645 |

Table 1 gives statistics on R2R, RxR, en-RxR and Landmark-RxR. The total number of sub-instructions from Landmark-RxR is 166,740, which contains 133,602 sub-instructions in train split, 13,591 sub-instructions in validation seen split, and 19,547 sub-instructions in validation unseen split. The average number of edges in a sub-trajectory is 1.6 with 21 words in the corresponding sub-instruction on average. The number of sub-trajectories contained in Landmark-RxR is 46,645. Landmark-RxR has the largest scale of instructions and trajectories and the minimum granularity among current human-annotated datasets. The fine-grained annotations are essential to effective cross-modal alignment learning and more accurate evaluation. In addition, because of the minimum granularity principle, the dataset can be easily expanded to a larger scale with different granularities by recombining the sub pairs.

## 4 Re-initialization Mechanism and Evaluation Metrics

### 4.1 Re-initialization Mechanism

An agent needs to navigate following the instructions strictly. So it is essential to evaluate the fidelity between predicted trajectories and the instructions exactly. Current metrics like Coverage weighted by Length Score (CLS) [16] and the normalized Dynamic Time Warping (nDTW) [17] try to obtain the fidelity by computing the similarity between predicted trajectory and the demonstration

---

[1] Note that the number of public instructions in RxR test split is 12,469.

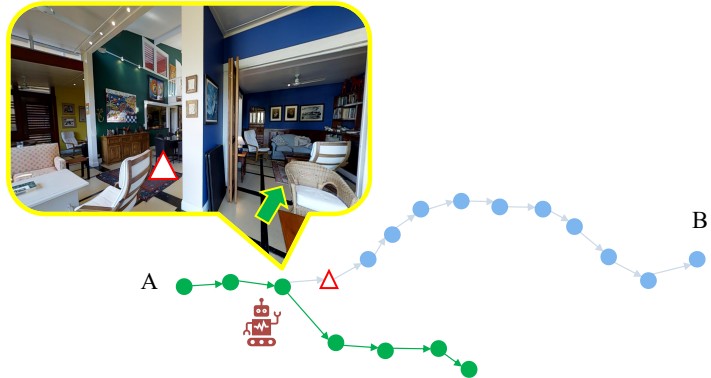

Figure 2: Agent deviates from the demonstration trajectory after encountering the difficult point. The green points indicate points in the predicted trajectory. The light blue points indicate points in the demonstration trajectory. The red triangle represents a difficult point in the demonstration trajectory.

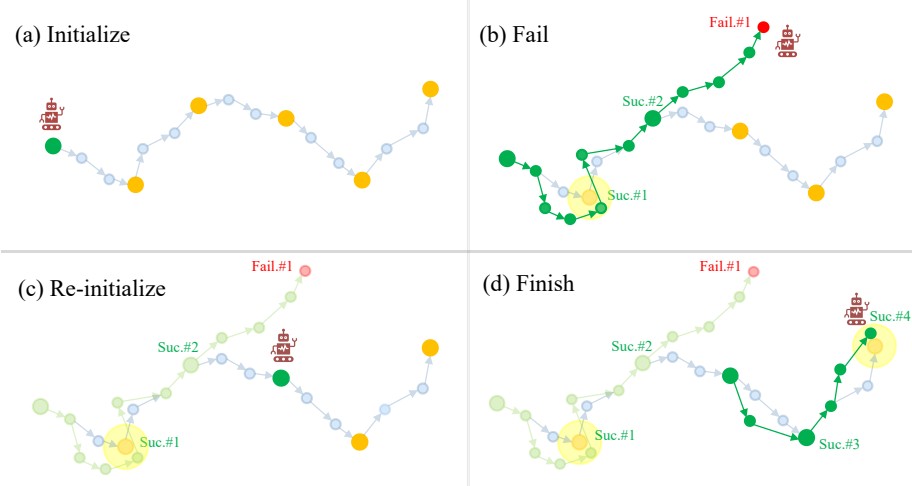

Figure 3: Re-initialization Mechanism. The green points mean the points in the predicted trajectory. The light green points mean the predicted trajectory before the last re-initialization and dark green points mean the predicted trajectory after the last re-initialization. The light blue points mean points in the demonstration trajectory. The orange points mean the landmarks in the demonstration trajectory. The red point represents the failure point where to re-initialize the agent.

trajectory. However, the nDTW and CLS scores are sensitive to the positions of difficult points, and have a deviation as a result when evaluating the navigation ability. As illustrated in Figure 2, the agent is initialized at point A and ready to follow the given instruction navigating to point B. Then during navigation, the agent deviates from demonstration trajectory seriously after encountering the difficult point and can only get a low fidelity score. The difficult point indicates the point where next action is hard to choose because of the difficult visual scene, *e.g.*, cluttered and similar objects in Figure 2, or ambiguous expressions in an instruction. However, for the same agent navigating from point B to point A, it is easy to finish the majority of the trajectory and get a higher fidelity score because the difficult point is just near the end. The distributions of difficult points in trajectory significantly impact current fidelity metrics, especially for the long trajectories.

Based on the landmark annotations in Landmark-RxR, we introduce the re-initialization mechanism to alleviate the influence of difficult points sensitivity of the fidelity metrics. We define a trajectory as $T = [ST_1, ST_2, ..., ST_{|T|}]$, where $ST_i = [P_{i1}, P_{i2}, ..., L_i]$, and corresponding instruction is defined as $I = [SI_1, SI_2, ..., SI_{|I|}]$. $P_{ij}$ means the $j^{th}$ point in the $i^{th}$ sub-trajectory $ST_i$. $L_i$ means the $i^{th}$ landmark in the trajectory, in other words, the end point of the $i^{th}$ sub-trajectory and $SI_i$ means the $i^{th}$ sub-instruction.

As illustrated in Figure 3, if the agent reaches within 3 meters, which is a fixed threshold for success metric defined in [4], of the landmark $L_i$ $(i < |T|)$, or the agent can stop within 3 meters of the landmark $L_{|T|}$ in $\pi$ ($\pi$ is set to 10 in this paper) steps from the last landmark, the navigation to landmark $L_i$ is successful. The failure is defined that the agent fails to get within 3 meters of the landmark $L_i$ $(i < |T|)$ in $\pi$ steps from the last landmark, or stops 3 meters away from the landmark $L_{|T|}$. If the agent fails in navigating to landmark $L_i$ $(i < |T|)$, we re-initialize the agent at landmark $L_i$ as the new start point, and reset the given instruction as $I_{reset} = [SI_{i+1}, ..., SI_{|I|}]$.

Since the re-initialization mechanism can correct serious deviations in time, for a complete trajectory, the averaged fidelity score of its sub-trajectories will be much less insensitive to the difficult points and can better reflect the navigation ability no matter where the difficult points locate.

## 4.2 Evaluation Metrics

An agent should have high navigation accuracy and robustness, which means its trajectories should match the instructions accurately and the agent should make as few mistakes as possible during navigation. Based on the re-initialization mechanism which could effectively alleviate the influence of difficult points when using fidelity metrics, Sub-Trajectory Accuracy (SA) and Success weighted by Sub-Trajectory Accuracy (SSA) are defined to measure the navigation accuracy, and Loss Number (LN) is defined to measure navigation robustness.

**Sub-Trajectory Accuracy**   This metric is defined as normalized Dynamic Time Warping (nDTW) [17] of the predicted sub-trajectory.

**Success weighted by Sub-Trajectory Accuracy**   This metric is defined as normalized Dynamic Time Warping (SDTW) [17] of the predicted sub-trajectory.

**Loss Number**   Every time agent fails to navigate to the next landmark, Loss Number will add one. The total numbers of landmarks in validation seen split and validation unseen split of Landmark-RxR are 13,591 and 19,547 respectively.

## 5   Our Method

We implement our baseline with an encoder-decoder architecture similar to the RCM [6] implemented in [14]. Please refer to supplemental material for more details about the baseline.

To enhance the local alignment between instructions and corresponding critical points, we propose two kinds of focal-oriented rewards benefiting from the fine-grained supervision in Landmark-RxR. The focal-oriented rewards emphasize the critical points in trajectories, which are more helpful for an agent to identify navigation directions because of their more detailed descriptions in instructions. Existing rewards are commonly based on either fidelity metric nDTW [2] or success metric SR. To fully demonstrate the key role of critical points in reward shaping, we choose to modify these two metrics to the focal-oriented rewards, namely soft focal-oriented reward and hard focal-oriented reward respectively. Since the landmarks in Landmark-RxR naturally meet the requirements of critical points, we sample the critical points for each trajectory from its landmark set annotated in our fine-grained Landmark-RxR.

**Soft focal-oriented reward**   In focal-oriented reward, all points in demonstration trajectories are taken into consideration. We base soft focal-oriented reward on the nDTW metric, which is used to identify the optimal alignment between predicted and demonstration trajectories in time order with minimized cumulative distance. Then we add an importance factor $\lambda$ to modify the nDTW metric. The importance factor $\lambda$ gives higher weight to the distances related to critical points, so it can bias the predicted trajectories in a soft manner, making them closer to the demonstration trajectories at positions of critical points especially.

$$R^{soft}_{focal}(R, Q) = exp(-\frac{min_W \Sigma_{(i_k, j_k) \in W} d(r_{i_k}, q_{j_k}) \cdot \lambda_{i_k}}{|R| \cdot d_{th}})$$ (1)

$$\lambda_{i_k} = \begin{cases} 10, & r_{i_k} \in S_R \\ 1, & others \end{cases}$$ (2)

---

[2] CLS is less used than nDTW because it is order-invariant and not ideal in some scenarios as described in [17].

where $W$ is a warping of the predicted trajectory and demonstration trajectory defined in [17], $d$ is a function that calculates distance between the two given points, $R = \{r_1, r_2, ..., r_{|R|}\}$ means the demonstration trajectory consisting of point $r_i$, $Q = \{q_1, q_2, ..., q_{|Q|}\}$ means the predicted trajectory consisting of point $q_i$ and $S_R = \{C_1, C_2, ..., C_{|S_R|}\}$ means the set of sampled critical point $C_i$ in demonstration trajectory. By contrast, the $\lambda$ is fixed to 1 in fidelity-oriented reward. So in the soft focal-oriented reward, the agent is penalized more once it fails to reach the critical points.

**Hard focal-oriented reward**  In hard focal-oriented reward, only the sampled critical points are considered in the final reward. If the agent succeeds in reaching the sampled critical points, the reward is positive to encourage a series of decisions during navigation. Otherwise, the reward is set as a negative value for penalization. The hard focal-oriented reward function is defined as:

$$R_{focal}^{hard}(R, Q) = \Sigma_{C \in S_R}\mathbb{I}(min_{q \in Q}d(q, C) < d_{th}) \tag{3}$$

$$\mathbb{I}(x) = \left\{ \begin{array}{ll} 1, & x \ is \ True \\ -1, & x \ is \ False \end{array} \right. \tag{4}$$

The hard focal-oriented reward encourages the agent to reach critical points during navigation and gives the agent more flexibility in exploring the trajectories between the critical points.

**Policy gradient**  We get loss from the non-differentiable reward as Equation 5:

$$L(\theta) = -\Sigma_t logp(a_t|s_t, \theta) \cdot (Re(a_t, s_t) - Re_{avg}) \tag{5}$$

where $a_t$ is the chosen action, $s_t$ is the visual and textual state, $\theta$ is the parameters of the network, $p$ is the probability of choosing the action $a_t$, $Re$ is the reward and $Re_{avg}$ is the averaged reward. Then the gradients to the network parameters are computed as:

$$\nabla_\theta L(\theta) = -\Sigma_t(Re(a_t, s_t) - Re_{avg})\nabla_\theta logp(a_t|s_t, \theta) \tag{6}$$

## 6 Experiment

### 6.1 Setup

**Data**  The training data involves four parts: the sub-instruction and sub-trajectory pairs (sub pairs) from Landmark-RxR, the synthesized instruction and synthesized trajectory pairs (synthesized pairs) which are augmented data obtained by concatenating several continuous sub pairs like [16], the complete instruction and complete trajectory pairs (complete pairs) from en-RxR, and the instruction and trajectory pairs from R2R. In validation phase, for en-RxR and R2R, the agent is validated based on their validation splits directly. For validation in Landmark-RxR, the agent navigates based on the complete instructions which is same to en-RxR when initialized in the beginning and the instructions synthesized by unfinished sub-instructions after re-initialization.

**Training policy**  We warm the agent up by imitation learning with student-forcing strategy. Then we switch to reinforcement learning to learn a more generalizable policy to unseen environments. During the reinforcement learning phase, we adopt the focal-oriented rewards and policy gradient [26] to update parameters in the network. Model training consumes about 1,600 minutes at the stage of imitation learning and 3,400 minutes at the stage of reinforcement learning on a single GTX3090 GPU.

**Hyper parameters**  We use the visual feature from the ResNet [27] trained on ImageNet [28] and adopt panoramic action space like [5]. The word embedding is initialized by GloVe300 [29]. The importance factor $\lambda$ in soft focal-oriented reward is set to 10. For the focal-oriented reward, we sample the same number of critical points in each trajectory to regularize the range of the reward value for each episode. $R_{focal}^{soft}$ and $R_{focal}^{hard}$ both sample 2 critical points from landmark set in each trajectory, which has the best trade-off between SR and Loss Number metrics empirically. The maximum navigation step $\pi$ allowed for each sub-trajectory is set to 10. The batch size is set to 100 and learning rate is $1 \times 10^{-4}$. The total iterations are 100,000 for imitation learning and 20,000 for reinforcement learning.

Table 2: Multitask [15] results on R2R validation split. *sub* means the sub pairs from Landmark-RxR. *co* means the complete pairs from en-RxR.

| # | R2R | *sub* | *co* | Validation Seen | | | Validation Unseen | | |
|---|---|---|---|---|---|---|---|---|---|
| | | | | nDTW↑ | SDTW↑ | SR↑ | nDTW↑ | SDTW↑ | SR↑ |
| 1 | √ | | | 56.7 | 44.0 | 53.9 | 31.2 | 19.1 | 33.0 |
| 2 | | √ | | 44.6 | 23.6 | 31.1 | 35.1 | 15.0 | 21.6 |
| 3 | | | √ | 49.7 | 34.0 | 43.7 | 32.4 | 15.8 | 23.6 |
| 4 | √ | | √ | 65.3 | 54.5 | 63.0 | 34.3 | 22.4 | 33.1 |
| 5 | √ | √ | √ | **65.7** | **55.6** | **66.6** | 39.5 | 27.0 | 38.4 |
| 6 | √ | √ | | 63.4 | 53.1 | 64.1 | **40.3** | **27.5** | **40.1** |

**Evaluation metrics**  To fully demonstrate the effectiveness of fine-grained data and the focal-oriented rewards, for R2R and en-RxR datasets, we adopt Success Rate (SR), the percentage of stopping within 3 meters of the final goal, normalized Dynamic Time Warping (nDTW), measuring the fidelity between predicted trajectory and the demonstration trajectory, Success weighted by normalized Dynamic Time Warping (SDTW), measuring the success rate and trajectory fidelity meanwhile. For Landmark-RxR dataset, we adopt Sub-Trajectory Accuracy (SA) and Success weighted by Sub-Trajectory Accuracy (SSA), difficult points insensitive metrics measuring the navigation accuracy, and Loss Number (LN), a difficult points insensitive metric measuring the navigation robustness.

## 6.2   Results and Analysis

**Fine-grained data contribute to instruction domain generalization**  Table 2 shows the performance of model#4 which is trained on R2R and the complete pairs of en-RxR is equivalent to model#1 which is only trained on R2R in unseen environments on SR metric. This indicates the complete pairs in en-RxR have little help for the navigation task in R2R because of the instruction domain gap, which is firstly found in [15]. Once we replace the complete pairs of en-RxR to sub pairs of Landmark-RxR as training data, the SR of model#6 improves 7.1% in unseen environments compared to model#1. The better instruction domain generalization indicates that the supervision from the fine-grained data do well with the domain of R2R. Since our Landmark-RxR can be extended to different granularities easily due to the minimum granularity principle, it can also benefit the generalization to different instruction domains.

**Fine-grained data contribute to unseen environments generalization**  As shown in Table 3, trained on synthesized pairs, model#8 outperforms model#7, model#9 and even model#12 in unseen environments on all metrics. It implies that fine-grained data with suitable granularity boost the performance in unseen environments most.

Table 3 and Table 4 show that, when we couple the complete pairs and sub pairs as training data (model#11), we see that SR improves 5.5%, and LN reduces 700 times with SSA improving 3.5% meanwhile compared to model#9. Trained on the joint granularity of sub pairs, synthetic pairs and complete pairs, model#13 outperforms models#7-12 on all metrics in unseen environments. These indicate cross-modal alignment supervision from coarse-grained data is not enough but the supervision from fine-grained data and coarse-grained data can complement each other to enhance the unseen environments generalization ability.

**Focal-oriented rewards facilitate local cross-modal alignment**  We incorporate different rewards in the reinforcement learning phase, with results on en-RxR and Landmark-RxR recorded in Table 3 and Table 4 respectively. Although model#14 which takes nDTW as reward has the highest fidelity scores in unseen environments, it gets the lowest SR and LN scores among all rewards. This indicates that only considering the global alignment between instructions and trajectories make agent just concern about the trajectories similarities but not the locations that instructions really concern during navigation.

Compared to model#14, model#17 (soft focal-oriented reward) also bases on nDTW but modifies it with more concern to the critical points. As illustrated in Table 3 and Table 4, the model#17 outperforms the model#14 by a significant margin with SR increasing from 29.3% to 33.7%, SSA increasing from 52.6 to 53.3 and LN decreasing from 5500 to 5173. In addition, model#17 also

Table 3: Results on en-RxR validation seen and validation unseen splits. $syn$ means the synthesized pairs augmented from Landmark-RxR.

| | IL Train | | | RL Train | | | | Validation Seen | | | Validation Unseen | | |
|---|---|---|---|---|---|---|---|---|---|---|---|---|---|
| # | $sub$ | $syn$ | $co$ | $R_{focal}^{soft}$ | $R_{focal}^{hard}$ | $R_{nDTW}$ | $R_{SR}$ | nDTW↑ | SDTW↑ | SR↑ | nDTW↑ | SDTW↑ | SR↑ |
| 7 | √ | | | | | | | 26.5 | 12.8 | 20.4 | 21.8 | 10.2 | 18.6 |
| 8 | | √ | | | | | | 41.7 | 29.0 | 42.2 | 24.6 | 13.8 | 24.2 |
| 9 | | | √ | | | | | 44.1 | 29.9 | 40.9 | 24.3 | 12.4 | 20.7 |
| 10 | √ | √ | | | | | | 42.1 | 30.1 | 43.8 | 24.9 | 14.9 | 26.1 |
| 11 | √ | | √ | | | | | 45.8 | 34.2 | 48.2 | 25.1 | 15.5 | 26.2 |
| 12 | | √ | √ | | | | | 47.6 | 35.3 | 48.6 | 23.9 | 13.6 | 22.8 |
| 13 | √ | √ | √ | | | | | 46.9 | 36.0 | 50.6 | 26.7 | 17.3 | 28.4 |
| 14 | √ | √ | √ | | | √ | | 55.0 | 32.8 | 40.7 | **43.2** | 22.6 | 29.3 |
| 15 | √ | √ | √ | | | | √ | 50.7 | 36.3 | 48.2 | 38.2 | 23.9 | 32.7 |
| 16 | √ | √ | √ | | | √ | √ | 54.4 | 39.3 | 49.9 | 41.0 | 25.0 | 33.6 |
| 17 | √ | √ | √ | √ | | | | **55.1** | **40.7** | **51.4** | 41.0 | **25.2** | 33.7 |
| 18 | √ | √ | √ | | √ | | | 48.5 | 37.0 | 50.7 | 36.6 | 24.2 | **34.9** |

Table 4: Results on Landmark-RxR validation seen and validation unseen splits. $syn$ means the synthesized pairs augmented from Landmark-RxR.

| | IL Train | | | RL Train | | | | Validation Seen | | | Validation Unseen | | |
|---|---|---|---|---|---|---|---|---|---|---|---|---|---|
| # | $sub$ | $syn$ | $co$ | $R_{focal}^{soft}$ | $R_{focal}^{hard}$ | $R_{nDTW}$ | $R_{SR}$ | SA↑ | SSA↑ | LN↓ | SA↑ | SSA↑ | LN↓ |
| 7 | √ | | | | | | | 56.6 | 51.7 | 4337 (31.9) | 49.4 | 44.1 | 7849 (40.2) |
| 8 | | √ | | | | | | 59.1 | 56.5 | 3267 (24.0) | 49.7 | 46.2 | 7029 (36.0) |
| 9 | | | √ | | | | | 57.6 | 54.5 | 3566 (26.2) | 48.0 | 43.8 | 7452 (38.1) |
| 10 | √ | √ | | | | | | 60.6 | 58.5 | 2899 (21.3) | 50.4 | 47.3 | 6677 (34.2) |
| 11 | √ | | √ | | | | | 60.9 | 58.6 | 2899 (21.3) | 50.5 | 47.1 | 6752 (34.5) |
| 12 | | √ | √ | | | | | 60.6 | 58.2 | 3039 (22.4) | 49.0 | 45.3 | 7302 (37.4) |
| 13 | √ | √ | √ | | | | | 61.8 | 59.9 | 2589 (19.0) | 51.7 | 48.9 | 6256 (32.0) |
| 14 | √ | √ | √ | | | √ | | 64.3 | 59.8 | 2764 (20.3) | **58.4** | 52.6 | 5500 (28.1) |
| 15 | √ | √ | √ | | | | √ | 61.6 | 58.8 | 2661 (19.6) | 55.5 | 51.7 | 5367 (27.5) |
| 16 | √ | √ | √ | | | √ | √ | 63.0 | 59.5 | 2622 (19.3) | 57.2 | 52.5 | 5309 (27.2) |
| 17 | √ | √ | √ | √ | | | | **63.8** | **60.7** | 2423 (17.8) | 57.6 | **53.3** | 5173 (26.5) |
| 18 | √ | √ | √ | | √ | | | 60.7 | 58.6 | **2360 (17.4)** | 54.7 | 51.8 | **4951 (25.3)** |

keeps a slight advantage over model#16 (fidelity-oriented reward[3]) with SSA increasing from 52.5 to 53.3 and LN down from 5309 to 5173, about a 0.7% drop of the total sub-trajectories number 19,547.

The soft focal-oriented reward is replaced by the hard form in model#18 (hard focal-oriented reward). Compared with the model#15 (goal-oriented reward) which also uses SR[4] as reward signal but only on the final goal point, SR of model#18 increases from 32.7% to 34.9% and LN of model#18 decreases 416 times meanwhile, 2.2% of the total number. Compared with model#16, model#18 has 1.3% increased in SR and 358 decreased in LN during navigation, a 1.9% drop of the total number 19,547. As illustrated in Figure 4, model#18 gives more attention to critical point related words like *close door*, and chooses the correct action, which indicates that model#18 performs local cross-modal alignment better. The above experiments suggest that our focal-oriented rewards which focus on local cross-modal alignment help the agent better understand what instructions concern and make fewer mistakes during navigation.

**Comparison with SoTA** As shown in Table 5, on the en-RxR benchmark, The model Ours$^{soft}$ which is trained on the data with joint granularities and equipped with soft focal-oriented reward outperforms RCM$^{rxr}$ on validation unseen split[5], with 4.9 and 8.1% improvement on SDTW and SR metrics respectively. In addition, the model Ours$^{hard}$ which is trained on the data with joint

---

[3] We formulate the fidelity-oriented reward based on nDTW as 'fidelity metric + SR', because experiment empirically show that it performs better on Loss Number metric than using the gain in nDTW score after taking an action as the reward signal like [17].

[4] Since trajectories in RxR are not necessary the global shortest to the goal, the reduced distance related part is not included in the goal-oriented reward here.

[5] Note that monolingual agent is not allowed to test on RxR competition platform. We compare our methods to the state-of-the-art method only on validation unseen split of en-RxR.

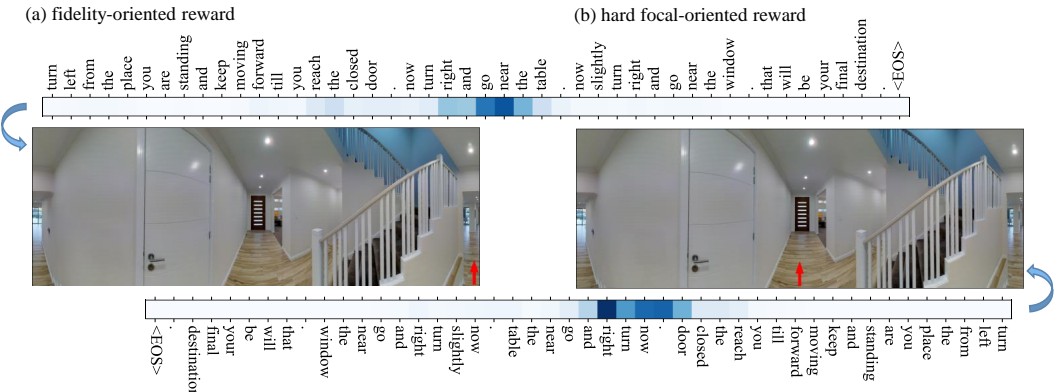

Figure 4: Compare agents trained on fidelity-oriented reward and hard focal-oriented reward. Agent trained on fidelity-oriented reward fails when choosing candidate action to close door, but agent trained on hard focal-oriented reward succeed to align the instruction to the right candidate action.

Table 5: Comparison of performance with the state-of-the-art methods on en-RxR. $RCM^{rxr}$ is the RCM[6] model trained on en-RxR implemented in [15]. $Ours^{soft}$ and $Ours^{hard}$ are the same models to model#17 and model#18 respectively.

| | Validation Unseen | | |
|---|---|---|---|
| Agent | nDTW↑ | SDTW↑ | SR↑ |
| $RCM^{rxr}$ [15] | **41.3** | 20.3 | 25.6 |
| $Ours^{soft}$ | 41.0 | **25.2** | 33.7 |
| $Ours^{hard}$ | 36.6 | 24.2 | **34.9** |

granularities and equipped with hard focal-oriented reward achieves the best result on SR metric with 34.9% on validation unseen split, 9.3% definite improvement compared to $RCM^{rxr}$.

# 7 Conclusion

Strong cross-modal alignment ability is the guarantee of successful navigation for a VLN agent. In this paper, we first proposed a human-annotated, fine-grained dataset Landmark-RxR. Our Landmark-RxR provides fine-grained supervision which can complement the weak cross-modal alignment supervision from coarse-grained data. Then benefiting from the fine-grained annotations, two kinds of focal-oriented rewards are proposed focusing on the local cross-modal alignment learning. In addition, the re-initialization mechanism is proposed for difficult points insensitive evaluation on Landmark-RxR. Experiments show that our fine-grained data and focal-oriented rewards help the agent gain superior navigation performance. However, this work is only a preliminary exploration based on the fine-grained Landmark-RxR dataset. We believe that there is much room for further investigation based on the Landmakr-RxR, like data augmentation, curriculum reinforcement learning and pretraining. In addition, our fine grain related work like focal-oriented rewards can not only enlightens for VLN task, but also for research on other tasks that can be modeled as Markov decision processes.

# 8 Acknowledgements

This work was jointly supported by National Key Research and Development Program of China Grant No. 2018AAA0100400, National Natural Science Foundation of China (61633021, 61721004, 61806194, U1803261, and 61976132), Beijing Nova Program (Z201100006820079), Shandong Provincial Key Research and Development Program (2019JZZY010119), Key Research Program of Frontier Sciences CAS Grant No.ZDBS-LY-JSC032, and CAS-AIR.

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
