# Landmark-RxR: Solving Vision-and-Language Navigation with Fine-Grained Alignment Supervision ——— Supplemental Material ———

Keji He[1,2]    Yan Huang[1,2]    Qi Wu[3]    Jianhua Yang[5]    Dong An[1,4]    Shuanglin Sima[1,2]
Liang Wang[*1,2,6,7]

[1]Center for Research on Intelligent Perception and Computing
National Laboratory of Pattern Recognition
Institute of Automation, Chinese Academy of Sciences
[2]School of Artificial Intelligence, University of Chinese Academy of Sciences
[3]School of Computer Science, University of Adelaide
[4]School of Future Technology, University of Chinese Academy of Sciences
[5]School of Artificial Intelligence, Beijing University of Posts and Telecommunications
[6]Center for Excellence in Brain Science and Intelligence Technology (CEBSIT)
[7]Chinese Academy of Sciences, Artificial Intelligence Research (CAS-AIR)
{keji.he, dong.an, shuanglin.sima}@cripac.ia.ac.cn    {yhuang,
wangliang}@nlpr.ia.ac.cn    qi.wu01@adelaide.edu.au    youngjianhua@bupt.edu.cn

## Appendix

## A  Baseline

Our baseline is similar to the RCM [1] implemented in [2] and is an encoder-decoder architecture. The words in instructions are encoded in reverse order in [2], and are encoded sequentially in our version.

The encoder includes an embedding layer and an LSTM layer.

$$\overline{w}_i = LSTM(embedding(w_i)), \quad i = 1, 2, ..., m \tag{1}$$

where $w_i$ is the tokenizer encoded $id$ of the $i^{th}$ word in the instruction, $m$ is the length of the instruction, $\overline{w}_i$ is the context-aware feature of $i^{th}$ word in the instruction.

The decoder includes a vision attention module, a text attention module and an action prediction module. At the $t^{th}$ step, the vision attention module computes attention weights for the $i^{th}$ view $v_{t,i}$ in panoramic representation and gets the attentive visual feature $\widetilde{v}_t$.

$$\widetilde{v}_t = \Sigma_i softmax_i(v_{t,i}^T W_v(\widetilde{h}_{t-1}^T W_{\widetilde{H}_1})^T) \cdot v_{t,i}, \quad i = 1, 2, ..., 36 \tag{2}$$

The agent randomly drops the concatenation of the attentive visual feature $\widetilde{v}_t$ and previous action feature $\widetilde{a}_{t-1}$ and then inputs them combined with the context-aware hidden state $\widetilde{h}_{t-1}$ to an LSTM and a dropout layer to get the hidden state $h_t$.

$$h_t = drop(LSTM(drop([\widetilde{v}_t; \widetilde{a}_{t-1}]), \widetilde{h}_{t-1})) \tag{3}$$

For the text attention module, analogous to the vision attention module, the agent computes the attentive textual feature $\widetilde{w}_t$.

$$\widetilde{w}_t = \Sigma_i softmax_i(\overline{w}_i^T(h_t^T W_H)^T) \cdot \overline{w}_i, \quad i = 1, 2, ..., m \tag{4}$$

---

[*]Corresponding author

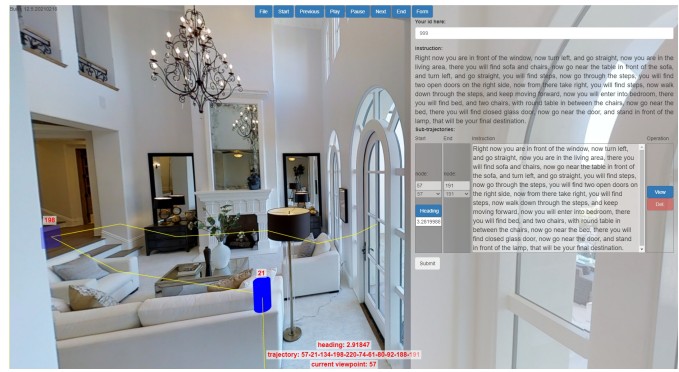

(a) Initialize at the start point (point#57).

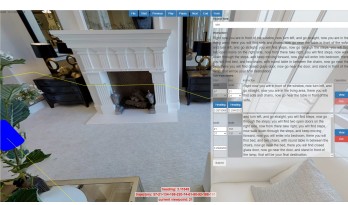

(b) Mark the $1^{st}$ landmark (point#21) and split out the corresponding sub-instruction.

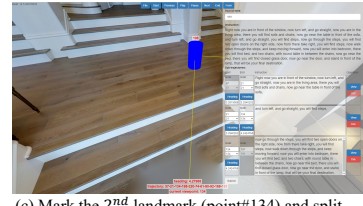

(c) Mark the $2^{nd}$ landmark (point#134) and split out the corresponding sub-instruction.

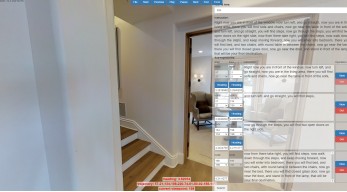

(d) Mark the $3^{rd}$ landmark (point#198) and split out the corresponding sub-instruction.

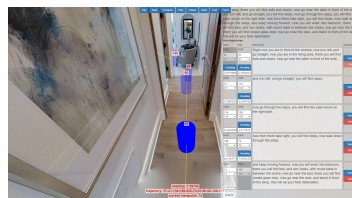

(e) Mark the $4^{th}$ landmark (point#74) and split out the corresponding sub-instruction.

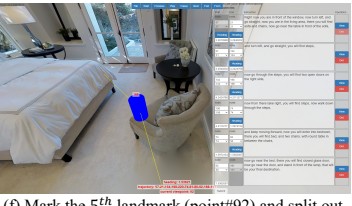

(f) Mark the $5^{th}$ landmark (point#92) and split out the corresponding sub-instruction.

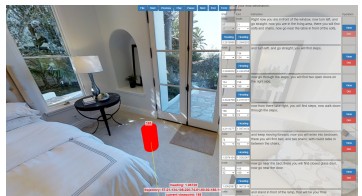

(g) Mark the $6^{th}$ landmark (point#188) and split out the corresponding sub-instruction.

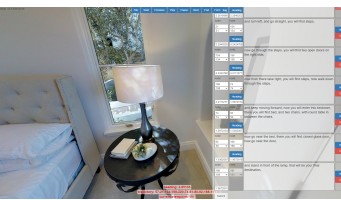

(h) Reach the end point (point#191).

Figure 1: Screenshots of an annotation process in our dataset collection tool.

To compute context aware hidden state $\widetilde{h}_t$, the agent performs linear and $tanh$ transformations on the concatenation of attentive textual feature $\widetilde{w}_t$ and the hidden state $h_t$.

$$\widetilde{h}_t = tanh([\widetilde{w}_t; h_t]^T W_1) \tag{5}$$

The probability $p(a_{t,k})$ moving to the $k^{th}$ candidate point is calculated as the softmax of the Hadamard product of the $k^{th}$ candidate viewpoint feature $c_{t,k}$ and the context aware hidden state $\widetilde{h}_t$.

$$p(a_{t,k}) = softmax_k(((c_{t,k}^T W_c) \circ (\widetilde{h}_t^T W_{\widetilde{H}_2})) W_2) \tag{6}$$

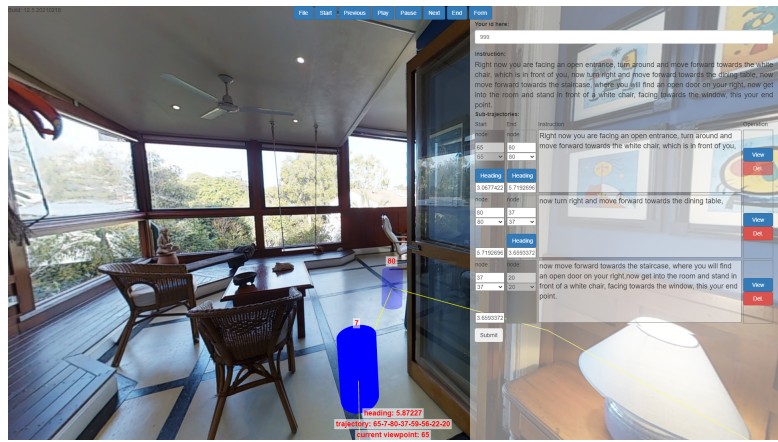

Figure 2: A screenshot of the verification process in our dataset collection tool.

## B  Data Collection

We show the screenshots of the annotation and verification processes with our web-based collection tool in Figure 1 and Figure 2. An example of sub-instruction and sub-trajectory pairs split from the complete instruction and complete trajectory pair in en-RxR [3] according to the five principles is shown in Figure 3.

## C  Fine-Grained Datasets

**Comparison with other fine-grained datasets**  Compared to our human-annotated Landmark-RxR which is based on en-RxR, Fine-Grained R2R [4] and BabyWalk [2] are based on R2R [5], whose trajectories are shorter and instructions are much simpler with fewer descriptions of the visual scenes. In addition, they generate sub-instruction and sub-trajectory pairs with heuristic rules, and this inevitably leads to inaccurate matches as illustrated in Figure 4. In terms of the dataset scale, the number of sub-instructions in Landmark-RxR exceeds the other two datasets by one order of magnitude. Table 1 shows our Landmark-RxR has the largest scale of fine-grained instructions and trajectories among all fine-grained datasets.

Table 1: Statistics on Fine-Grained R2R, BabyWalk, and our Landmark-RxR. Edge means the number of edges each trajectory contains on average.

| Dataset | Edge | Instruction | | | | | Trajectory |
| --- | --- | --- | --- | --- | --- | --- | --- |
| | | Train | Validation Seen | Validation Unseen | Test | Total | |
| Fine-Grained R2R [4] | 1.5 | 51,377 | 3,775 | 8,481 | 15,385 | 79,018 | 34,602 |
| BabyWalk [2] | 2.8 | 25,551 | 1,863 | 4,189 | 7,646 | 39,249 | 19,895 |
| Landmark-RxR | 1.6 | 133,602 | 13,591 | 19,547 | - | 166,740 | 46,645 |

**Quality of fine-grained data from Landmark-RxR**  In our comparison experiments, the trajectories from en-RxR are divided into the same number of sub-trajectories as Landmark-RxR equally or randomly, and the corresponding sub-instructions are obtained according to the word-level alignment annotations in en-RxR. The resulting datasets are called Eq-RxR and Rand-RxR. Table 2 shows

Table 2: Performances of models trained on Eq-RxR, Rand-RxR and Landmark-RxR.

| # | Dataset based | en-RxR | | | Landmark-RxR | | |
| --- | --- | --- | --- | --- | --- | --- | --- |
| | | nDTW↑ | SDTW↑ | SR↑ | SA↑ | SSA↑ | LN↓ |
| 1 | Eq-RxR | 19.7 | 8.5 | 15.8 | 47.0 | 40.3 | 8923 (45.6) |
| 2 | Rand-RxR | 19.0 | 8.6 | 17.0 | 47.8 | 42.8 | 8353 (42.7) |
| 3 | Landmark-RxR | **21.8** | **10.2** | **18.6** | **49.4** | **44.1** | **7849 (40.2)** |

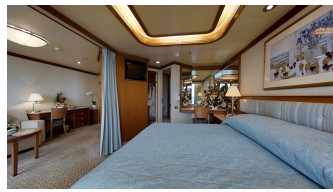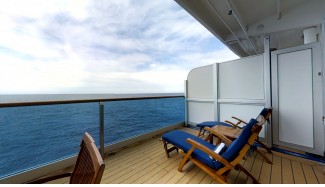

You are standing in a bedroom, facing towards a bed, now turn to your left, in front of you you can see a glass door, open the glass door and walk out of the room. Now to the left you can see a round table with four chairs around it, to the right you can see relaxing chairs.

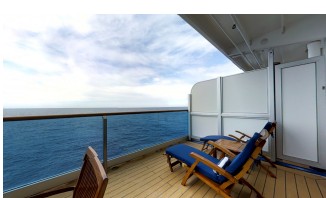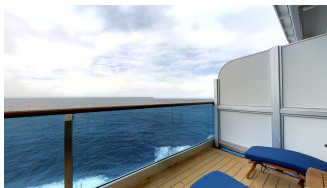

Walk to the relaxing chairs. Now in front of you you can see a white cart board wall.

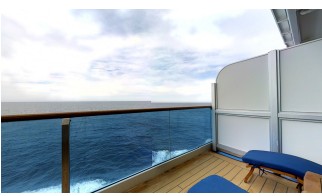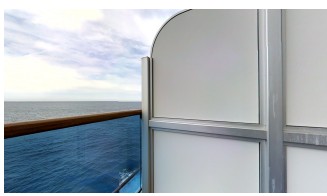

Walk to the wall.

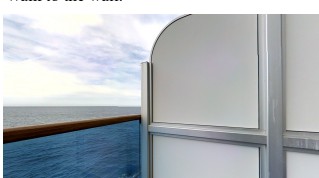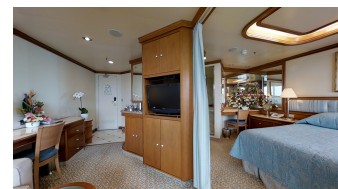

Now turn to your right, in front of you you can see another glass door, open the glass door and enter the room. To the left you can see tea-poi with some fruits in a bowl on it.

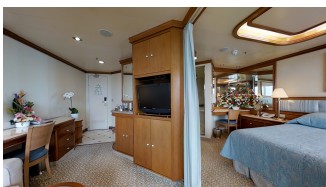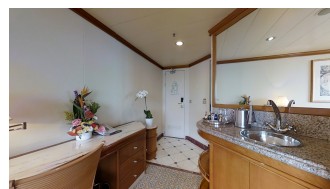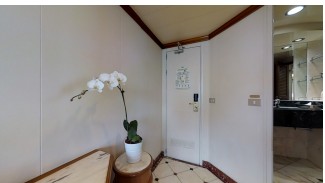

Walk straight, now on the right you have an sink, to the left you have an table with chair in-front of it, beside the table you can see a stand with flower vase on it, walk to the stand. That's your end point.

Figure 3: Sub-instruction and sub-trajectory pairs split from a complete instruction and complete trajectory pair in en-RxR.

the performances of different models which are trained on the Eq-RxR, Rand-RxR and Landmark-RxR respectively. The model trained on Landmark-RxR outperforms other two models in all metrics on the unseen validation splits of both en-RxR and Landmark-RxR, demonstrating the high quality of Landmark-RxR annotations.

(a) Mismatches in Fine-Grained R2R.

(b) Mismatches in BabyWalk.

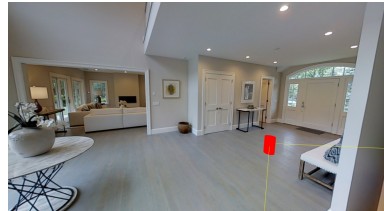

right at the round table.

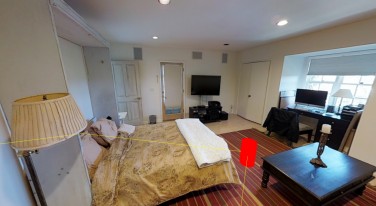

walk around the bed and exit the room
going straight to the end of the hallway.

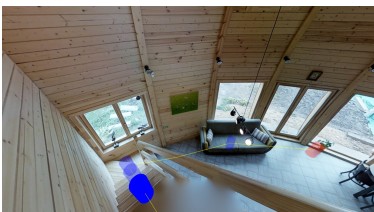

go down to the bottom of the stair.

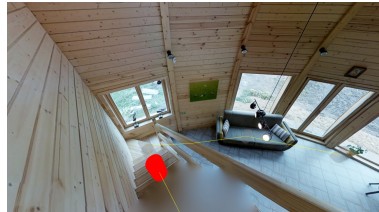

turn around and walk down the stairs
to the bottom.

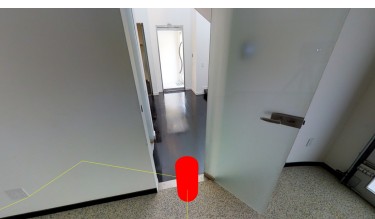

and proceed down the step.

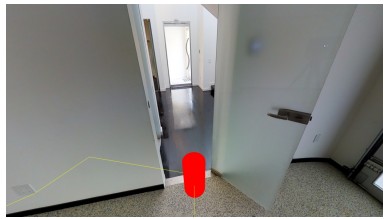

exit garage and turn left and walk
down stairs.

Figure 4: Examples of mismatches between sub-instructions and sub-trajectories in Fine-Grained R2R and BabyWalk. The red points indicate end points of the sub-trajectories.

# D  Datasheet: Landmark-RxR

The following datasheet documents dataset Landmark-RxR as [6].

## Motivation

**For what purpose was the dataset created?**
Landmark-RxR is created for fine-grained learning and evaluation in VLN task. It can provide fine-grained supervision that complements the coarse-grained supervision for both imitation and reinforcement learning phases. In addition, it can support fine-grained evaluation like the re-initialization mechanism proposed in this paper.

## Composition

**What do the instances that comprise the dataset represent (e.g., documents, photos, people, countries)?**
The dataset Landmark-RxR includes fine-grained sub-instruction and the sub-trajectory pairs split from en-RxR dataset [3].

**How many instances are there in total (of each type, if appropriate)?**
Landmark-RxR contains 166,740 sub-instructions and 46,645 sub-trajectories. For more details, please refer to Appendix C.

**Does the dataset contain all possible instances or is it a sample (not necessarily random) of instances from a larger set?**

Our Landmark-RxR is based on dataset en-RxR. We split all the instruction and trajectory pairs from en-RxR and get the corresponding sub-instruction and sub-trajectory pairs included in our Landmark-RxR.

**What data does each instance consist of?**
Each instance contains one sub-instruction and one sub-trajectory represented as a series of points. The examples of sub-instruction and sub-trajectory pairs can be seen in Figure 3.

**Is there a label or target associated with each instance?**
Points of sub-trajectories are the labels in the training phase.

**Is any information missing from individual instances?**
No.

**Are relationships between individual instances made explicit (e.g., users movie ratings, social network links)?**
Yes. We totally have 90 buildings and each of them has a unique identifier. All the sub-trajectories are sampled from the buildings.

**Are there recommended data splits (e.g., training, development/validation, testing)?**
Yes. Our Landmark-RxR is based on en-RxR and shares the same building splits to en-RxR. In addition, please refer to Section 3.2 in the main paper for more details about the data splits.

**Are there any errors, sources of noise, or redundancies in the dataset?**
During the annotation process, some errors in en-RxR have been corrected in our Landmark-RxR. In addition, although some errors will inevitably occur in data collection, we have tried to reduce the error rate as much as possible. The Landmark-RxR is manually annotated following the five principles strictly and has been verified again after the annotations are finished.

**Is the dataset self-contained, or does it link to or otherwise rely on external resources (e.g., websites, tweets, other datasets)?**
The Landmark-RxR is based on dataset en-RxR. We split the instruction and trajectory pairs from en-RxR and get the corresponding sub-instruction and sub-trajectory pairs included in our Landmark-RxR.

**Does the dataset contain data that might be considered confidential (e.g., data that is protected by legal privilege or by doctor-patient confidentiality, data that includes the content of individuals non-public communications)?**
No.

**Does the dataset contain data that, if viewed directly, might be offensive, insulting, threatening, or might otherwise cause anxiety?**
No.

**Does the dataset identify any subpopulations (e.g., by age, gender)?**
No.

**Is it possible to identify individuals (i.e., one or more natural persons), either directly or indirectly (i.e., in combination with other data) from the dataset?**
No.

**Does the dataset contain data that might be considered sensitive in any way (e.g., data that reveals racial or ethnic origins, sexual orientations, religious beliefs, political opinions or union memberships, or locations; financial or health data; biometric or genetic data; forms of government identification, such as social security numbers; criminal history)?**
No.

## Collection

**How was the data associated with each instance acquired?**
Please refer to Section 3.2 in the main paper for details about the annotation process.

**Over what timeframe was the data collected?**
From January 2021 to May 2021.

**What mechanisms or procedures were used to collect the data (e.g., hardware apparatus or sensor, manual human curation, software program, software API)?**
A 3D web-based collection tool is designed for annotation. Please refer to Section 3 in the main paper and the screenshots of annotation and verification processes in Figure 1 and Figure 2 in the supplemental material respectively.

**If the dataset is a sample from a larger set, what was the sampling strategy (e.g., deterministic, probabilistic with specific sampling probabilities)?**
We obtain fine-grained annotations based on all data in en-RxR.

**Who was involved in the data collection process (e.g., students, crowdworkers, contractors)?**
Totally 30 annotators participated in the annotation task, contributing about 2,700 hours.

**Does the dataset relate to people?**
No.

**Did you collect the data from the individuals in question directly, or obtain it via third parties or other sources (e.g., websites)?**
We collect the data directly from the individuals.

**Were the individuals in question notified about the data collection?**
Yes.

**Did the individuals in question consent to the collection and use of their data?**
Yes.

## Preprocessing / Cleaning / Labeling

**Was any preprocessing/cleaning/labeling of the data done(e.g.,discretization or bucketing, tokenization, part-of-speech tagging, SIFT feature extraction, removal of instances, processing of missing values)?**
During the annotation process, some errors in en-RxR have been corrected in our Landmark-RxR.

**Was the raw data saved in addition to the preprocessed/cleaned/labeled data (e.g., to support unanticipated future uses)?**
Yes.

## Uses

**Has the dataset been used for any tasks already?**
Yes. We use the dataset for the VLN task in both imitation learning and reinforcement learning.

**Is there a repository that links to any or all papers or systems that use the dataset?**
No.

**What (other) tasks could the dataset be used for?**
Similar cross-modal tasks, like image/video caption, visual question answer and cross-modal retrieval can all use our dataset for better cross-modal alignment learning. In addition, back-translation, reward shaping and pretraining can also base on our dataset.

**Is there anything about the composition of the dataset or the way it was collected and prepro-cessed/cleaned/labeled that might impact future uses?**
No.

## Distribution

**Will the dataset be distributed to third parties outside of the entity (e.g., company, institution, organization) on behalf of which the dataset was created?**
Yes. The dataset will be distributed to the research community.

**How will the dataset will be distributed (e.g., tarball on website, API, GitHub)?**
GitHub.

**When will the dataset be distributed?**
We will release our dataset around December.

**Will the dataset be distributed under a copyright or other intellectual property (IP) license, and/or under applicable terms of use (ToU)?**
Landmark-RxR is distributed under a CC-BY license.

**Have any third parties imposed IP-based or other restrictions on the data associated with the instances?**
Yes. Anyone who uses the Matterport3D should follow the MATTERPORT END USER LICENSE AGREEMENT. In addition, data from en-RxR are under the CC-BY license.

## Maintenance

**Will the dataset be updated (e.g., to correct labeling errors, add new instances, delete in-stances)?**
There may be minor changes for future research requirements.