# OpenReview forum: "Landmark-RxR: Solving Vision-and-Language Navigation with Fine-Grained Alignment Supervision"
_NeurIPS.cc/2021/Conference — NeurIPS 2021 Poster_

### Official Review · Reviewer_gPfu · 2021-07-06

**Rating:** 6
**Confidence:** 3

**Summary:**

The paper tackles the problem of Vision and Language Navigation (VLN). The main contribution of the paper is a dataset called Landmark RxR. The instructions in the RxR dataset have been broken down into smaller segments by human annotators, where each segment corresponds to navigation to a landmark. A landmark is a point at which it is difficult to make a decision (due to ambiguity, clutter, etc). The paper proposes a "re-initialization" strategy to better evaluate VLN agents. Two types of reward functions based on the deviation from the landmarks are defined. The paper reports results on R2R, en-RxR and Landmark RxR datasets.

**Limitations And Societal Impact:**

No, the paper does not address the limitations or the potential negative societal impact.

**Main Review:**

**Strengths**

- The fine-grained annotation of the RxR dataset can be useful in certain navigation scenarios.

- The paper shows state-of-the-art results on the en-RxR dataset.

- One of the proposed reward functions (Soft focal-oriented reward) provides significant improvements compared to standard goal-oriented or fidelity-oriented reward functions.

**Weaknesses**

One major issue is that it is not clear if the improvements are provided by the landmark annotations or just by breaking the long trajectories. Two baseline experiments should be added to the experiment section: (1) divide the instructions into equal size segments (2) divide the instructions into segments of random size (ending at a period). Without these two experiments, it is hard to judge if the provided dataset is a useful contribution or not.

Also, there are several critical points that are not clear:

- Is the re-initialization mechanism used only for computing SA and SSA metrics? or is it used for computing the other metrics as well?

- The performance drops when complete instructions are used in Table 2 (model #5 vs model #6), but using complete instructions provides improvement in Table 3 (line 266). What is the explanation for that?

- Why does the method use only two points among the landmarks (line 250)?

- Regarding Table 2, which dataset has been used for rows that do not have a tick for R2R?

I will adjust my rating based on the rebuttal. There are also numerous typos.

**Post-rebuttal comments**

The rebuttal addresses my concerns (the baseline experiments and the response to the questions) so I retain my initial rating.

**Time Spent Reviewing:**

6

---

> ### Author Response · Authors · 2021-08-10
> **Author Responses to Reviewer gPfu**
>
> **Q1: One major issue is that it is not clear if the improvements are provided by the landmark annotations or just by breaking the long trajectories. Two baseline experiments should be added to the experiment section: (1) divide the instructions into equal size segments (2) divide the instructions into segments of random size (ending at a period). Without these two experiments, it is hard to judge if the provided dataset is a useful contribution or not.**
>
> A1:
>
> |           | |en-RxR|||Lk-RxR| |
> | --------- | ------ | ------ | ------ |  ------ | ------ | ------ |
> | #         | nDTW↑  | sDTW↑ | SR↑  | SA↑ | SSA↑| LN↓ |
> | ours         | 21.8    | 10.2 | 18.6  | 49.4 | 44.1 | 7849 (40.2) |
> | baseline1 | 19.7    | 8.5   | 15.8  | 47.0 | 40.3 | 8923 (45.6) |
> | baseline2 | 19.0    | 8.6   | 17.0  | 47.8 | 42.8 | 8353 (42.7) |
>
> 1. Thanks for your insightful suggestion. We divide the complete trajectory and instruction pairs from en-RxR into equal size and random size segments to train #baseline1 and #baseline2 models separately. Our model (#ours) is only trained on sub-instruction and sub-trajectory pairs from the Landmark-RxR, the same as #model7 in Table3. Three models are tested on the unseen validation split of en-RxR and Landmark-RxR (LK-RxR).
> The experimental results are reported in the table above. Our model outperforms both the #baseline1 and #baseline2 models on all metrics significantly, with 2.8% and 1.6% improved on SR and 4.4% and 2.5% dropped on Loss Number (LN) separately. The results indicate that our Landmark-RxR has high-quality annotations and has a useful contribution to the community.
>
> 2. Except as training data, our Landmark-RxR is also useful in the validation phase to better evaluate the navigation model, for example, in the way of the proposed re-initialization mechanism.
>
> **Q2: Is the re-initialization mechanism used only for computing SA and SSA metrics? or is it used for computing the other metrics as well?**
>
> A2: In addition to SA and SSA, the re-initialization mechanism is also used to compute the Loss Number metric. As mentioned on Lines 177-178, every time the agent fails to navigate to the next landmark, the agent will be re-initialized at the next landmark and the Loss Number adds one.
>
> **Q3: The performance drops when complete instructions are used in Table 2 (model #5 vs model #6), but using complete instructions provides improvement in Table 3 (line 266). What is the explanation for that?**
>
> A3: This is a good point. It is mainly because of the domain difference which is first found in [1]. Compared with R2R whose trajectories are short and instructions are simple, the much longer paths and richer instructions from en-RxR are out-of-domain. So when the complete instruction and trajectory pairs (complete pairs) from en-RxR are used as training data, it biases the model away from the domain of R2R and leads to the inferior performance on R2R. The performance on en-RxR and Landmark-RxR reported in Table3 is significantly improved by complete pairs, because the training data and validation data are both based on en-RxR and share the same domain.
>
> **Q4: Why does the method use only two points among the landmarks (line 250)?**
>
> A4: Empirically, we find that the model performs best when only two points are sampled from the landmark set. We will add more discussions about this in the revision.
>
> **Q5: Regarding Table 2, which dataset has been used for rows that do not have a tick for R2R?**
>
> A5: For Table 2, all rows use R2R for validation. For row 2 and row 3 that do not have a tick for R2R, row 2 uses sub-instruction and sub-trajectory pairs from Landmark-RxR as training data and row 3 uses complete instruction and trajectory pairs from en-RxR as training data. We will refine our descriptions.
>
> [1] Alexander Ku, Peter Anderson, Roma Patel, Eugene Ie, and Jason Baldridge. Room-across-room: Multilingual vision-and-language navigation with dense spatiotemporal grounding. Empirical Methods in Natural Language Processing, 2020.

---

### Official Review · Reviewer_dQiv · 2021-07-11

**Rating:** 5
**Confidence:** 3

**Summary:**

The paper proposes to improve vision and language navigation by augmenting the RxR dataset with corresponded subinstructions and subtrajectories.  Using the notion of critical points (points that are more important/helpful in the navigation process), two kinds of focal-oriented rewards (soft and hard) are proposed to encourage the predicted trajectory to be close to the demonstration trajectories at critical points.  The authors conduct experiments on R2R and RxR and show that using the collected subinstructions and subtrajectories and proposed losses, the trained agent is able to outperform the baseline RCM model.  For the RxR dataset, the training data consisted both of original RxR data as well as an augmented dataset consisting of concatenating the collected subinstruction and subtrajectories together.

Main contributions
- Landmark-RxR: Additional annotations on top of the RxR dataset providing sub-instructions paired with sub-trajectories
- Proposed to use modified rewards for improved cross-modal alignment
- Ablation experiments of proposed method on the R2R and RxR dataset

**Ethical Concerns:**

No ethical concerns.

**Limitations And Societal Impact:**

There was no discussion of the limitations of the work or the any negative social impacts.

**Main Review:**

Originality: Limited.  Much of what is done in this work has been proposed in prior work (having paired sub-instruction and sub-trajectory data, concatenating simpler instructions/trajectories together, the focus-oriented reward is also similar to rewards from prior work).  The main novelty lies in the application sub-instructions to the RxR dataset, using the sub-instructions to generate more novel combinations, and having the reward to restricted to a set of critical points.

Quality: There are missing discussion and comparisons wrt prior work.  There is no discussion on the limitations and weaknesses of the work.

Clarity: The writing is hard to understand at places.  The paper is also missing some important details (see "Other comments and questions" for details)

Significance: The Landmark-RxR dataset will be useful for the community.  The significance of the proposed method is unclear.

Strengths:
- The collected Landmark-RxR dataset would be valuable
- Experiments show that using the subinstructions and the proposed focal losses improves over the baseline RCM model

Weaknesses:
- Missing detailed discussion and comparison against related work (Jain et al, 2019, Ilharco et al, 2019 both used fidelity-oriented rewards with RCM).  There should be more detailed description of how this work relates and clear experiments comparing against the prior work.
- Weak experimental setup.
  - Missing comparisons against relevant prior work.  This includes the close works of RCM with fidelity oriented reward [Jain et al, 2019, Ilharco et al, 2019], as well as other prior work using sub-instructions (such as BabyWalk [Zhu et al, 2020], [Hong et al 2020]).
  - Proposed method appears to considerably underperform prior work with RCM (Wang et al, 2019, Jain et al, 2019, Ilharco et al, 2019)
  - For Table 4 (RxR against SOTA), results are reported only for on set (not all sets of validation seen, unseen, testset).  It's also not clear which set Table 4 numbers corresponds to (Table 4 indicates validation seen, but the test (lines 295) imply validation unseen - actual numbers appear to match val-unseen).
- There is no discussion of training variance
- Contribution of the method is unclear
- The writing is also poor, making it difficult to understand at places.  There are also terminology/wording that are not properly introduced and thus unclear.

Other comments and questions:

- What is the relationship between critical points and landmarks?

- When is the re-initialization mechanism used?  Is it only for evaluation?  Or is it used during training? inference?

- Is there a reason why the loss number metric is not normalized against the number of landmarks?  The lack of normalization makes it challenging to intuitively determine how good the agent is without some understanding of the total number of landmarks.

- It's unclear how the sub-trajectory accuracy is computed.  Is this an average of the nDTW of all sub-trajectories?  If it is a nDTW, then the name "sub-trajectory accuracy" does not seem accurate (wouldn't "sub-trajectory nDTW" be better?)

- How does the proposed soft and hard focal-oriented reward relate to the fidelity-based reward from Jain et al, 2019?

- Line 201: What specifically is the distance function used? The L2 Euclidean distance?
- Line 204: How are the critical points sampled? Is a random set of the landmarks uniformly sampled?

- In the experiments, the R_nDTW and R_SR rewards are not fully explained.  How do these rewards correspond to what was introduced in prior work?  The combination of R_nDTW and R_SR seem to work well (comparable with the proposed method).

- The subinstructions/subtrajectories are concatenated together for data augmentation.  Was there experiments comparing it against using data augmentation with the speaker-follower model?

- Line 254: The claim that performance of model#1 and model#4 are equivalent is inaccurate.  While they have similar SR for val-unseen, the performance of the two methods are clearly different for other metrics

- Overall the writing quality was poor, making it confusing and hard to understand.  See below for details:
  - From the introduction, it was hard to understand what exactly the dataset consisted of and what specifically was meant by various terms such as "fine-grained perspective" and "landmarks".  Overall, I found the term "landmark" to be very confusing as I would expect a "landmark" (for a indoor scene) to be say a "bed" or a "painting on the wall" and correspond to either visual landmarks or landmarks identified by text.  But what it seems to refer to is a location on the trajectory (defined in the paper as "marked end points of sub-trajectories").  The term "critical point" was also not precisely defined.   Would all landmarks qualify to be a "critical point"?

  - Line 57-58: "english Guide part" - the details of the RxR dataset has not been introduced so it is hard to understand what "english Guide" refers to. It's also unclear what "fine-grained" and "currently the largest scale" means.

  - The set of annotation principles should be described in the main paper, not the supplemental.  Overall, the description in the main paper was not very informative.  The information in supplemental is clearer and more informative.  In addition to the annotation principles, the supplemental also explains how it differs from prior fine-grained dataset in a clear way, which was lacking from the main paper.  Table 1 from the supplement should be in the main paper (Table 1 in the main paper is better suited to be in the supplement)

  - Line 245: It would make more sense if the "Hyper parameters" were described after the "Training policy" and before the "Evaluation metrics"

  - Some other example of phrases with poor wording that are difficult to understand:
    - Line 38: "no sub-instruction can we get from RxR"
    - Line 85: "to navigate in recurrent mechanism like LSTM efficiently"
    - Line 95: "spatially-temporally aligned priori"
    - Line 96: "a plain result"
    - Line 183: "in reverse order originally"
    - Line 230: "The agent reasons in one stage and end-to-end manner"

  - Some examples of wording / typos
    - Line 3, 24, and others: "prediction trajectory" => "predicted trajectory"
    - Line 98: "Besides," => "In addition,"
    - Line 105: "supports" => "allows"
    - Line 106-107: "During the annotation progress" => "During annotation"
    - Line 111: "firstly" => ""
    - Figure 2 caption: "mean" => "indicate"
    - Line 157: "succeeds to reach within 3 meters" => "reaches within 3 meters"
    - Line 142: "firstly" => ""
    - Line 219: "the data contain two parts" => "the training data consists of two parts"
    - Table 2 caption: "have little help" => "provides little improvement"

   - References are not alphabetically sorted, references not also not capitalized correctly

Post-Rebuttal:
- I increased my rating from 4 to 5.  The clarification by the authors has mitigated some of my concerns.  However, I still believe that the exposition lacking in clarity and detail, with terminology that is not clearly defined.   Due to the contribution of the dataset and the authors clarifications, I increase my score to 5.   I believe the current submission is not clear enough and a revision is needed.

**Time Spent Reviewing:**

6

---

> ### Author Response · Authors · 2021-08-10
> **Author Responses to Reviewer dQiv (Part 2)**
>
> **Q13: It's unclear how the sub-trajectory accuracy is computed. Is this an average of the nDTW of all sub-trajectories? If it is a nDTW, then the name "sub-trajectory accuracy" does not seem accurate (wouldn't "sub-trajectory nDTW" be better?)**
>
> A13: As shown on Line 172, Sub-Trajectory Accuracy (SSA) is defined as the nDTW of the predicted sub-trajectory. It is reported in Table 3 as the average of the nDTW of all sub-trajectories. We call it Sub-Trajectory Accuracy because we use it to measure the navigation accuracy. We will follow your great suggestion to improve its name.
>
>
> **Q14: How does the proposed soft and hard focal-oriented reward relate to the fidelity-based reward from Jain et al, 2019?**
>
> A14: The proposed soft and hard focal-oriented rewards have no direct relationship with the fidelity-oriented reward [3]. Because the CLS [3] metric is order-invariant, we only choose nDTW [4] as the fidelity metric to design the fidelity-oriented reward (model#16 in Table 3) in this paper.
>
>
> **Q15: What specifically is the distance function used? The L2 Euclidean distance?**
>
> A15: Same as defined in the nDTW [4] metric, we use the shortest distance between two points in the simulated environment (Matterport3D [7]) as the distance function.
>
> **Q16: How are the critical points sampled? Is a random set of the landmarks uniformly sampled?**
>
> A16: For a trajectory with $n$ landmarks, we choose the $\llcorner n/2 \lrcorner_{th}$ landmark and the last landmark as the sampled critical points.
>
> **Q17: In the experiments, the R_nDTW and R_SR rewards are not fully explained. How do these rewards correspond to what was introduced in prior work? The combination of R_nDTW and R_SR seem to work well (comparable with the proposed method).**
>
> A17:
> 1. As shown on Lines 273-274, the R_nDTW (model# 14) reward takes the nDTW metric as a reward directly. It is the original form before we modify it to the soft focal-oriented reward. No prior work has only used it as a reward. We report R_nDTW in Table 3 to conclude that only considering the global alignment between instructions and trajectories makes the agent just concern about the similarity of trajectories but not the locations that instructions really concern during navigation.
> 2. As shown in the caption of Table 3 and Line 285, the R_SR reward is the goal-oriented reward [5] which uses the SR metric as a reward signal.
> 3. As mentioned in the caption of Table 3, the combination of R_nDTW and R_SR is exactly the fidelity-oriented reward [4].
>
> **Q18: The subinstructions/subtrajectories are concatenated together for data augmentation. Was there experiments comparing it against using data augmentation with the speaker-follower model?**
>
> A18: The reason why we concatenate sub-instructions/sub-trajectories together is to verify the conclusions in this paper but not for data augmentation:
> * Fine-grained data with suitable granularity boost the performance. (Lines 262-263)
> * Fine-grained and coarse-grained data can complement each other to enhance the unseen environment generalization ability. (Lines 269-271)
>
> Therefore, this paper has no experiment comparing the concatenation method against the speaker model [8]. In addition, we point out that data augmentation is one of the future investigation directions based on our Landmark-RxR dataset in the Conclusion part. Furthermore, we believe that the fine-grained data from our Landmark-RxR are also helpful to the training of the speaker model for generating better pseudo instructions.
>
> **Q19: The claim that performance of model#1 and model#4 are equivalent is inaccurate. While they have similar SR for val-unseen, the performance of the two methods are clearly different for other metrics**
>
> A19:  Thanks for the reminder. We will refine the expression in the revision.
>
> **Q20: From the introduction, it was hard to understand what exactly the dataset consisted of and what specifically was meant by various terms such as "fine-grained perspective" and "landmarks". Overall, I found the term "landmark" to be very confusing as I would expect a "landmark" (for a indoor scene) to be say a "bed" or a "painting on the wall" and correspond to either visual landmarks or landmarks identified by text. But what it seems to refer to is a location on the trajectory (defined in the paper as "marked end points of sub-trajectories"). The term "critical point" was also not precisely defined. Would all landmarks qualify to be a "critical point"?.**
>
> A20:  Thank you for the reminder. We will improve our definitions in the revision.
> 1. Our Landmark-RxR dataset includes sub-instruction and sub-trajectory pairs split from instruction in en-RxR (Lines 103-104). We also discuss what is included in our dataset with Figure 1 (Lines 116-117) and given some examples in Figure 3 in the supplementary material.
> 2. "Fine-grained perspective" means we solve vision-and-language navigation with fine grain related methods, like our fine-grained dataset, focal-oriented rewards and re-initialization mechanism.
> 3. “Landmarks” means the end points of the sub-trajectories as defined in the paper (Lines 36). Since end points of the sub-trajectories (landmarks) follow the “Clear End Point Principle” (Part B in supplementary material), the landmarks are just the points that are clearly described in the instructions.
> 4. “Critical point” means the more valuable points for navigation because of their detailed descriptions in instructions (Lines 186-187).
> 5. All landmarks naturally meet the requirements of critical points and are qualified to be  “critical points”, because landmark annotations follow “Clear End Point Principle” which requires them to be described clearly in instructions.
>
> **Q21: Line 57-58: "english Guide part" - the details of the RxR dataset has not been introduced so it is hard to understand what "english Guide" refers to. It's also unclear what "fine-grained" and "currently the largest scale" means.**
>
> A21:
> 1. The following explanations are summarized from [6]. The RxR dataset includes a Guide split and a Follower split. The instructions of the Guide split are multilingual and the Follower split has no instruction. “English Guide part of RxR” means the part of the Guide split whose instructions are in English.
> 2. "Fine-grained data" means sub-instruction and sub-trajectory pairs with the same definition as [1, 2].
> 3. “Currently the largest scale” means our Landmark-RxR has the largest number of trajectories and instructions among all current VLN datasets.
>
> We will add a more detailed introduction of these concepts to our revision.
>
> **Q22: The set of annotation principles should be described in the main paper, not the supplemental. Overall, the description in the main paper was not very informative. The information in supplemental is clearer and more informative. In addition to the annotation principles, the supplemental also explains how it differs from prior fine-grained dataset in a clear way, which was lacking from the main paper. Table 1 from the supplement should be in the main paper (Table 1 in the main paper is better suited to be in the supplement)**
>
> A22: Thanks for your suggestion. We will follow them to improve our manuscript.
>
> **Q23: Line 245: It would make more sense if the "Hyper parameters" were described after the "Training policy" and before the "Evaluation metrics"**
>
> A23: Thanks for your suggestion. We will refine it in the revision.
>
> **Q24: Some other example of phrases with poor wording that are difficult to understand and wording / typos.**
>
> A24: Thanks for these suggestions, and we will fix them in the revision.
>
> [1] Yicong Hong, Cristian Rodriguez-Opazo, Qi Wu, and Stephen Gould. Sub-instruction aware vision-and-language navigation. Empirical Methods in Natural Language Processing, 2020.
> [2] Wang Zhu, Hexiang Hu, Jiacheng Chen, Zhiwei Deng, Vihan Jain, Eugene Ie, and Fei Sha. Babywalk: Going farther in vision-and-language navigation by taking baby steps. Association for Computational Linguistics, 2020.
> [3] Vihan Jain, Gabriel Magalhaes, Alexander Ku, Ashish Vaswani, Eugene Ie, and Jason Baldridge. Stay on the path: Instruction fidelity in vision-and-language navigation. Association for Computational Linguistics, 2019.
> [4] Gabriel Ilharco, Vihan Jain, Alexander Ku, Eugene Ie, and Jason Baldridge. General evaluation for instruction conditioned navigation using dynamic time warping. NeurIPS Visually Grounded Interaction and Language Workshop, 2019.
> [5] Xin Wang, Qiuyuan Huang, Asli Celikyilmaz, Jianfeng Gao, Dinghan Shen, Yuan-Fang Wang, William Yang Wang, and Lei Zhang. Reinforced cross-modal matching and self-supervised imitation learning for vision-language navigation. In IEEE Conference on Computer Vision and Pattern Recognition, pages 6629–6638, 2019.
> [6] Alexander Ku, Peter Anderson, Roma Patel, Eugene Ie, and Jason Baldridge. Room-across-room: Multilingual vision-and-language navigation with dense spatiotemporal grounding. Empirical Methods in Natural Language Processing, 2020.
> [7] Peter Anderson, Qi Wu, Damien Teney, Jake Bruce, Mark Johnson, Niko Sünderhauf, Ian Reid, Stephen Gould, and Anton Van Den Hengel. Vision-and-language navigation: Interpreting visually-grounded navigation instructions in real environments. In IEEE Conference on Computer Vision and Pattern Recognition, 2018.
> [8] Daniel Fried, Ronghang Hu, Volkan Cirik, Anna Rohrbach, Jacob Andreas, Louis-Philippe Morency, Taylor Berg-Kirkpatrick, Kate Saenko, Dan Klein, and Trevor Darrell. Speaker-follower models for vision-and-language navigation. Advances in Neural Information Processing Systems, 2018.

---

> > ### Comment · Reviewer_dQiv · 2021-08-24
> > **Thank you for your clarifications**
> >
> >
> > Thanks for providing the detailed response and the clarifications.  To clarify points of my review, I believe there is significant value in the novelty and contribution of the RxR landmark dataset.  I also believe that there is value in the proposed method and experiments.  However, there are issues with clarity of the writing and overall presentation (and I feel my complaints about the missing discussion/comparison again prior work is also mostly due to this clarity of writing) which should be improved.
> >
> > While the clarification addresses most of my concerns, there are some remaining points that could be clarified.
> > - Q4-6,14
> >   For Table 3, it was not clear exactly which rows correspond to the prior work ([3] (Stay on the Path, Jain et al 2019), [4] (Ilharco et al, 2019), [5] (Wang et al, 2019)).
> >   - It was clear whether it is indeed the case that goal-oriented reward (#model15) is equivalent to [5] and the fidelity oriented rewards(#model16) are equivalent to [3,4] or are there differences in the exact formulation of the loss for the goal/fidelity oriented rewards or are they exactly the same as implemented in prior work?
> >   - I think the response to Q4 indicates that [3] should be CLS+SR and does not correspond to any rows (thank  you for the clarification), while #model15 is equivalent to [5] and #model16 is equivalent to [4]?  Is this the case?  If so it would be good to clearly indicate that in the paper.  If not, differences should be pointed out.  Ideally, [3] should be added as a comparison as well.
> >
> > - Q10:
> >   I'm still not sure what makes a critical point "critical" and why are they sampled.
> >   Here is line 192-193:
> >   "Since the landmarks in Landmark-RxR naturally meet the requirements of critical points, we just sample the critical points for each trajectory from its landmark set annotated in our fine-grained Landmark-RxR."
> >   - This informs me that all landmarks are critical points.
> >   - If all landmark are critical points, why are the "landmarks" not all considered "critical points".  Why take a subset and call them "critical points"?  Why not just have a subset of "landmarks"?
> >   - Why is 2 critical points sampled?  Is that using all the landmarks?  If so, is it possible the other points are not as critical?
> >   - Also, what properties specifically define a "critical" point
> >   From earlier: line 49-52:
> >   "However, for points in trajectory, some critical ones are more helpful to navigation process, i.e., the intermediate points with more detailed description in the instruction can assist agent in correcting its trajectory in time and reaching the locations that the instruction really concern.  Thus, an emphasis on alignment between words in critical parts of the instructions and corresponding critical points in trajectories, which is in an local to local way, is essential."
> >   - I tried very hard to understand to understand exactly what properties "critical" points should have.  Is it that if the agent does not go to the "critical" point, the agent is no longer following the instructions?  Are the the same as "diifcult" points (described in Figure 2 and use in the re-initialization mechanism)?
> >
> > - Q13: To clarify with the sub-trajectory accuracy, as it is the nDTW of the predicted subtrajectory, is it the case that it then is equivalent to just considering subtrajectories as full trajectories?  So it ignores the original grouping of sub-trajectories into full trajectories?  My original question was whether if you had a full trajectory that consisted of 2 sub-trajectories, whether you would first take the average of the sub-trajectory sDTW for each trajectory and then get an average over the trajectories for the dataset, or you just take the average over all the sub-trajectories over the dataset.
> >
> > - Q15: How is the shortest distance between two points in the simulated environment computed?  Is it the L2 Euclidean distance?

---

> > > ### Author Response · Authors · 2021-08-27
> > > **Author Responses to Reviewer dQiv (Part 3)**
> > >
> > > **Q25: [Q4-6,14] For Table 3, which rows correspond to the prior work ([3] (Stay on the Path, Jain et al 2019), [4] (Ilharco et al, 2019), [5] (Wang et al, 2019))?  Are they exactly the same as implemented in prior work? If so it would be good to clearly indicate that in the paper. If not, differences should be pointed out. Ideally, [3] should be added as a comparison as well.**
> > >
> > > A25:
> > > 1. In Table 3, [5] (goal-oriented reward) and [4] (fidelity-oriented reward based on nDTW) corresponds to model#15 and model#16 respectively. The [3] (fidelity-oriented reward based on CLS) does not correspond to any rows.
> > > 2. In addition, there are a few differences between our reproduced methods and prior works. [4] proposes the nDTW to replace the CLS metric and uses the gain in nDTW score after taking an action as the reward signal, but our experimental results show that this reward type does not perform well on Loss Number metric (LN: 5426). So we formulate the fidelity-oriented reward based on nDTW as 'fidelity metric + SR' (model#16, LN: 5309), which is an effective formation for fidelity-oriented rewards that was first defined in [3]. For the goal-oriented reward, since trajectories in the R2R dataset are all the shortest trajectories to the goals, [5] use the reduced distance after taking an action as an extra reward signal in addition to the SR signal. However, 44.5% of RxR trajectories are not the shortest trajectories from the start to the goal location as described in the Path Statistic section of [6]. So we did not include the reduced distance part in the goal-oriented reward as mentioned in footnote 1 (page 8) in the paper.
> > >
> > > Thanks for your suggestions. We will add more details about differences and comparisons in the revision.
> > >
> > > **Q26: [Q10] What makes a critical point "critical" and what properties "critical" points should have?**
> > >
> > > A26:
> > > In the VLN task [7], the agent can only stand on the navigable points in the simulated environment (those blue dots in Fig 1) and move between them according to its decisions based on given instructions. For a trajectory corresponding to the given instruction, some points are described simply, but some are described in great detail. These points have different importance to the VLN task. For example, when the agent is asked to follow the instruction "keep going until you reach the vase", it needs to pay more attention to the point near the vase but not the midway points which have almost no corresponding description in the instruction. More attention to the point near the vase helps the agent align the visual object better with the instruction. By comparison, more attention to other points will make the agent learn the wrong cross-modal alignment because there is no description about them in the instruction.
> > >
> > > In summary, the points described clearly in the instructions are more important to the VLN task than others and we call them critical points. The detailed description in the instruction is the property of the critical point and makes a critical point "critical".
> > >
> > > **Q27: [Q10] Why are critical points sampled? why are the "landmarks" not all considered "critical points"? Why are 2 critical points sampled? Is it possible the other points are not as critical?**
> > >
> > > A27:
> > > All the landmarks are qualified to be critical points, so we sample critical points used in focal-oriented rewards from the landmark set. The focal-oriented rewards are designed to address the local cross-modal alignment problem and only choose two points but not all the points from the landmark set in a trajectory as critical points. It does not mean that unselected landmarks are not qualified to be the critical points. Attending to more sampled intermediate critical points can help the agent have better local alignment ability (with dropped Loss Number metric) in our experiments, but it makes the agent pay relatively less attention to navigating to the global goal locations (with dropped SR metric). Our experiments have empirically demonstrated that the focal-oriented rewards have the best performance (in terms of best balancing the Loss Number metric and SR metric) with two sampled critical points. So we set the hyper-parameter as 2.
> > > Thanks for your and Reviewer gPfu's reminder. We will add more discussion in our revision.
> > >
> > > **Q28: [Q13] To clarify with the sub-trajectory accuracy, as it is the nDTW of the predicted subtrajectory, is it the case that it then is equivalent to just considering subtrajectories as full trajectories? So it ignores the original grouping of sub-trajectories into full trajectories?**
> > >
> > > A28:
> > > When we compute the Sub-Trajectory Accuracy (SA), we just considering sub-trajectories like complete trajectories. For example, in Fig 3(d) where the agent is evaluated on the Landmark-RxR based on the re-initialization mechanism, we split the predicted trajectories (the light green trajectory and the heavy green trajectory) into five sub-trajectories according to the landmarks. For each sub-trajectory, we compare it with the corresponding groundtruth sub-trajectory to compute the SA just as we do for the complete trajectory when evaluating on the R2R/RxR datasets. It is worth mentioning that although we compute the SA independently for each sub-trajectory, the agent does not navigate each sub-trajectory separately during navigation but navigates the current unfinished part of the complete instruction according to the re-initialization mechanism.
> > > We will refine the expression in the revision and publish our codes including the computation of all the metrics on Landmark-RxR.
> > >
> > > **Q29: [Q13] My original question was whether if you had a full trajectory that consisted of 2 sub-trajectories, whether you would first take the average of the sub-trajectory sDTW for each trajectory and then get an average over the trajectories for the dataset, or you just take the average over all the sub-trajectories over the dataset.**
> > >
> > > A29:
> > > For the original question, our answer is: we first take the average of the sub-trajectory nDTW (Sub-Trajectory Accuracy, SA) and the sub-trajectory sDTW (Success weighted by Sub-Trajectory Accuracy, SSA) for a complete trajectory which can be written as SA-avg and SSA-avg. Then we average the SA-avg and SSA-avg of each complete trajectory in the Landmark-RxR dataset and report them in Table 3.
> > >
> > > **Q30: [Q15] How is the shortest distance between two points in the simulated environment computed? Is it the L2 Euclidean distance?**
> > >
> > > A30:
> > > It is not the L2 Euclidean distance. The distance means the path length between two points. Since the navigation graph (including points and edges) is available from the simulated environment, the shortest distance (namely the shortest path length) can be computed by the shortest path finding algorithms like the Dijkstra algorithm.
> > >
> > > [1] Yicong Hong, Cristian Rodriguez-Opazo, Qi Wu, and Stephen Gould. Sub-instruction aware vision-and-language navigation. Empirical Methods in Natural Language Processing, 2020.
> > > [2] Wang Zhu, Hexiang Hu, Jiacheng Chen, Zhiwei Deng, Vihan Jain, Eugene Ie, and Fei Sha. Babywalk: Going farther in vision-and-language navigation by taking baby steps. Association for Computational Linguistics, 2020.
> > > [3] Vihan Jain, Gabriel Magalhaes, Alexander Ku, Ashish Vaswani, Eugene Ie, and Jason Baldridge. Stay on the path: Instruction fidelity in vision-and-language navigation. Association for Computational Linguistics, 2019.
> > > [4] Gabriel Ilharco, Vihan Jain, Alexander Ku, Eugene Ie, and Jason Baldridge. General evaluation for instruction conditioned navigation using dynamic time warping. NeurIPS Visually Grounded Interaction and Language Workshop, 2019.
> > > [5] Xin Wang, Qiuyuan Huang, Asli Celikyilmaz, Jianfeng Gao, Dinghan Shen, Yuan-Fang Wang, William Yang Wang, and Lei Zhang. Reinforced cross-modal matching and self-supervised imitation learning for vision-language navigation. In IEEE Conference on Computer Vision and Pattern Recognition, pages 6629–6638, 2019.
> > > [6] Alexander Ku, Peter Anderson, Roma Patel, Eugene Ie, and Jason Baldridge. Room-across-room: Multilingual vision-and-language navigation with dense spatiotemporal grounding. Empirical Methods in Natural Language Processing, 2020.
> > > [7] Peter Anderson, Qi Wu, Damien Teney, Jake Bruce, Mark Johnson, Niko Sünderhauf, Ian Reid, Stephen Gould, and Anton Van Den Hengel. Vision-and-language navigation: Interpreting visually-grounded navigation instructions in real environments. In IEEE Conference on Computer Vision and Pattern Recognition, 2018.
> > > [8] Daniel Fried, Ronghang Hu, Volkan Cirik, Anna Rohrbach, Jacob Andreas, Louis-Philippe Morency, Taylor Berg-Kirkpatrick, Kate Saenko, Dan Klein, and Trevor Darrell. Speaker-follower models for vision-and-language navigation. Advances in Neural Information Processing Systems, 2018.

---

> > > > ### Comment · Reviewer_dQiv · 2021-09-03
> > > > **Thank you for the clarifications**
> > > >
> > > > Thank you for the further clarifications.  I still believe the writing and clarity of the submission need to be improved.  Due to the usefulness of the dataset and the clarifications, I have increased my score to 5.

---

> > > ### Author Response · Authors · 2021-09-02
> > > **Author Responses to Reviewer dQiv (Part 4)**
> > >
> > > Dear Reviewer dQiv,
> > >
> > > Thanks a lot for your efforts in reviewing this paper. We have properly addressed all your concerns and provided clarifications on all confusing concepts. Could you please kindly re-evaluate our paper based on the current situation? If you have any further questions, we are also very glad to discuss them.
> > >
> > > Thanks,
> > >
> > > Authors

---

> ### Author Response · Authors · 2021-08-10
> **Author Responses to Reviewer dQiv (Part 1)**
>
> **Q1: (Originality) Limited. Much of what is done in this work has been proposed in prior work (having paired sub-instruction and sub-trajectory data, concatenating simpler instructions/trajectories together, the focus-oriented reward is also similar to rewards from prior work). The main novelty lies in the application sub-instructions to the RxR dataset, using the sub-instructions to generate more novel combinations, and having the reward to restricted to a set of critical points.**
>
> A1: Thank you for such comprehensive comments. We would like to clarify the major concern of limited contribution from the following aspects.
> 1. We propose the current largest-scale, human-annotated sub-instruction dataset. In you mentioned related works, their used sub-instructions are automatically obtained by heuristic rules, which are not precise enough and limit the navigation performance. Reviewers JseM&Hcah also recognize this contribution by commenting that “it is the first paper to create sub-goal level human annotations for sub-instructions for the instructions in RxR”, and “makes meaningful contributions in terms of adding more fine-grained data to RxR dataset”.
> In addition, our experiments about fine-grained data are designed to demonstrate that the supervision from fine-grained and coarse-grained data can complement each other to improve the cross-modal alignment ability of the model itself. By contrast, although Hong et al. [1] and  Zhu et al. [2] also use fine-grained data, both of them use the fine-grained data mainly for a “one by one” strategy but not to investigate the relationship between different granularity data and improve the cross-modal alignment ability of the navigation model itself. During validation, they segment a given instruction into several sub-instructions that are much easier to navigate and mainly focus on how to navigate these easy sub-instructions one by one for better performance.
> 2. In addition to the dataset, we also make two key contributions in the model aspect.  1) We propose the focal-oriented rewards that focus on addressing the local cross-modal alignment problem with fine-grained supervision, while previous ones [3, 4, 5]  only consider global cross-modal alignment.  2) We also propose the re-initialization mechanism to fully evaluate the navigation process in a way that makes fidelity metric insensitive to difficult points.
> 3. Both the dataset and rewards can better deal with the cross-modal alignment problem during the navigation, and can help agents to generalize much better to instructions with domain gap by improving 8.4% on sDTW and 6.9% on SR and to unseen environments by improving 4.9% on sDTW and 8.1% on SR.
>
> **Q2: (Quality) There are missing discussion and comparisons wrt prior work.**
>
> A2:
> 1. As mentioned in the Introduction and Related Work section, we discuss prior works about cross-modal alignment, reward shaping and their disadvantages.
> 2. As mentioned in Section 3.2 and Part C in the supplementary material, we compare our Landmark-RxR with coarse-grained and fine-grained datasets.
> 3. As mentioned on Lines 96-97, we point out that our work focuses on using fine-grained supervision to benefit the cross-modal alignment ability.
> 4. As mentioned in Section 6.2, we compare our focal-oriented rewards with the goal-oriented and fidelity-oriented rewards in Table 3 and analyze the results.
>
> We are sorry Reviewer dQiv missed these details that have been presented in the original submission. We will highlight them in the revision.
>
> **Q3: (Quality) There is no discussion on the limitations and weaknesses of the work.**
>
> A3: Our focal-oriented rewards currently need annotated landmarks for critical points sampling. However, this is a too expensive way. In future work, we will investigate a learning way to sample the critical points.  We will give a clearer explanation of the limitations in the revision.
>
>
> **Q4: Missing detailed discussion and comparison against related work (Jain et al, 2019, Ilharco et al, 2019 both used fidelity-oriented rewards with RCM). There should be more detailed description of how this work relates and clear experiments comparing against the prior work.**
>
> A4: The major differences between our work and you mentioned previous ones [3, 4] are as follows.
> Our focal-oriented rewards focus on addressing the local cross-modal alignment problem with fine-grained supervision, while prior works only pay attention to the global cross-modal alignment, like the global goal points (goal-oriented reward [5]) and global trajectory similarity (fidelity-oriented reward [3, 4]).
> In addition, we have made the desired comparisons in Table 3 (model#16). For the fidelity-oriented reward, it can be decomposed as: fidelity metric + SR. In our experiment using model#16, we choose the nDTW [4] as the fidelity metric to design the fidelity-oriented reward. We compare our two kinds of focal-oriented rewards with the fidelity-oriented reward (model#16, nDTW+SR) in Table 3 and analyzed the results in Section 6.2. Our soft focal-oriented reward outperforms nDTW+SR with 0.7% dropped on Loss Number (LN), and our hard focal-oriented reward outperforms nDTW+SR with 1.8% dropped on LN and 1.3% improved on SR. In addition, we did not consider the CLS [3] metric in the paper because it is order-invariant and not ideal in some scenarios as described in [4]. For your reference, our results using CLS+SR as the fidelity-oriented reward on Landmark-RxR (Val Unseen) are SA (56.4), SSA(32.3), LN (5279) and on en-RxR (Val Unseen) are nDTW (39.6), sDTW (24.5), SR (32.9). Our soft focal-oriented reward outperforms CLS+SR with 0.5% dropped on LN and 0.8% improved on SR, and our hard focal-oriented reward outperforms CLS+SR with 1.7% dropped on LN and 2% improved on SR.
> We will add more discussion in the revision.
>
>
> **Q5: Missing comparisons against relevant prior work. This includes the close works of RCM with fidelity oriented reward [Jain et al, 2019, Ilharco et al, 2019], as well as other prior work using sub-instructions (such as BabyWalk [Zhu et al, 2020], [Hong et al 2020]).**
>
> A5:
> 1. Please refer to A4 for the comparison against the fidelity-oriented rewards [3, 4].
> 2. Please refer to the first point of A1 for the comparison against prior work using sub-instructions [1, 2].
>
> **Q6: Proposed method appears to considerably underperform prior work with RCM (Wang et al, 2019, Jain et al, 2019, Ilharco et al, 2019)**
>
> A6: Our proposed method can effectively improve the local cross-modal alignment ability and outperform prior works [3, 4, 5]. All these works only reported their results on a different dataset (R2R). On Landmark-RxR and en-RxR datasets, we have reproduced the goal-oriented reward [5] and fidelity-oriented reward [4], and compared our method with them under a fair setting.
> As illustrated in Table 3, our soft focal-oriented reward (model#17) outperforms the goal-oriented reward with 1.0% dropped on Loss Number and 1.0% improved on SR, and outperforms the fidelity-oriented reward with 0.7% dropped on Loss Number. Our hard focal-oriented reward outperforms the goal-oriented reward with 2.1% dropped on Loss Number and 2.2% improved on SR and outperforms the fidelity-oriented reward with 1.8% dropped on Loss Number and 1.2% improved on SR. The significant drop on the Loss Number metric indicates that the focal-oriented rewards make the model have a better local cross-modal alignment ability and make fewer mistakes during navigation.
>
>
> **Q7: 3For Table 4 (RxR against SOTA), results are reported only for on set (not all sets of validation seen, unseen, testset). It's also not clear which set Table 4 numbers corresponds to (Table 4 indicates validation seen, but the test (lines 295) imply validation unseen - actual numbers appear to match val-unseen).**
>
> A7: Thanks for pointing out this typo. Table 4 column heading should be “Validation Unseen”. We will correct it in the revision.
> Since only the performance on “Validation Unseen” of the state-of-the-art method  ($RCM^{rxr}$ [6]  in Table 4) is public and the monolingual model is not allowed to test on RxR competition platform, we compare our methods to the state-of-the-art method only on the unseen validation split of en-RxR.
>
>
> **Q8: There is no discussion of training variance**
>
> A8: Compared with the fidelity-oriented and goal-oriented rewards [4, 5], the training process of our focal-oriented rewards like the training variance, maintains very similar characteristics because we only modify the constant coefficients based on these rewards. Empirically, our method has a stable advantage over other methods. Thanks for your reminder. We will give a clearer explanation in the revision.
>
>
> **Q9:  The writing is also poor, making it difficult to understand at places. There are also terminology/wording that are not properly introduced and thus unclear.**
>
> A9: We apologize for the unclear parts. We will refine our paper.
>
>
> **Q10: What is the relationship between critical points and landmarks?**
>
> A10: As shown on Lines 192-193, we sample the critical points for each trajectory from its landmark set annotated in our fine-grained Landmark-RxR.
>
>
> **Q11: When is the re-initialization mechanism used? Is it only for evaluation? Or is it used during training? inference?**
>
> A11: As mentioned on Line 65, the re-initialization mechanism is only used for evaluation.
>
>
> **Q12: Is there a reason why the loss number metric is not normalized against the number of landmarks? The lack of normalization makes it challenging to intuitively determine how good the agent is without some understanding of the total number of landmarks.**
>
> A12: We normalize it in the Result&Analysis section but report it directly in Table 3 as it is defined. We apologize for this and will make it clearer in the revision.

---

### Official Review · Reviewer_HCah · 2021-07-16

**Rating:** 6
**Confidence:** 4

**Summary:**

This paper proposes augmenting the en-RxR dataset by splitting the trajectories into finer grained sub-trajectories (by human annotators). It also introduces soft and hard focal reward formulations for their policy gradient agent which seem to do well empirically.

**Limitations And Societal Impact:**

No negative societal impact of this work

**Main Review:**

Observing that cross-modal alignment is an import aspect leading to higher performance in VLN metrics, this paper makes meaningful contributions both in terms of adding more fine-grained data to RxR dataset and exploiting this additional signal to improve performance on the original task. I have a few suggestions below:

Suggestions

- Can you split Table 3 and report numbers for en-RxR and Landmark-RxR separately?

- Presentation
  - Remove all occurrences of "obviously"
  - There are several typos throughout the manuscript, please correct those

- Related work section missing relevant citations:

  - Transferable representation learning in vision and-language navigation. Huang et al
  - Multi-modal discriminative model for vision-and-language navigation. Huang et al
  - Touchdown: Natural Language Navigation and Spatial Reasoning in Visual Street Environments. Chen et al
  - Retouchdown: Adding Touchdown to StreetLearn as a Shareable Resource for Language Grounding Tasks in Street View. Mehta et al

**Time Spent Reviewing:**

3

---

> ### Author Response · Authors · 2021-08-10
> **Author Responses to Reviewer HCah**
>
> **Q1: Can you split Table 3 and report numbers for en-RxR and Landmark-RxR separately?**
>
> A1: Thank you for the suggestion. We will modify it to make it clearer in the reversion.
>
> **Q2: Presentation**
> 1. Remove all occurrences of "obviously"
> 2. There are several typos throughout the manuscript, please correct those
>
> A2: We will improve our presentation and correct all the typos in the revision.
>
> **Q3: Related work section missing relevant citations:**
>
> A3: We will cite and discuss these missing references in the revision.

---

### Official Review · Reviewer_JseM · 2021-07-18

**Rating:** 7
**Confidence:** 4

**Summary:**

This paper introduces a new fine-grained VLN dataset called Landmark-RxR which builds on top of RxR. Using the fine-grained annotations, they propose two kinds of focal-oriented rewards for improving local cross-modal alignment. They also propose a re-initialization mechanism to evaluate the navigation such that it is insensitive to difficult points.

**Main Review:**

### Originality and Significance:
-	Previous papers in VLN space have tried to split the instructions to sub-instructions using heuristics, but it is the first paper to create sub-goal level human annotations for sub-instructions for the instructions in RxR. This dataset will provide more fine-grained supervision for cross-modal alignment training.
-	Through their experiments, they support the claims that they make.
    -	Fine-grained data contribute to instruction domain generalization: They show a 7.1% increase in performance when using R2R data and sub-instructions from Landmark-RxR as opposed to just using R2R instructions or R2R+RxR instructions.
    -	Comparison with SoTA: Their model with soft focal-oriented rewards outperforms the RCM model by 4.9 in sDTW and 8.1% on SR.


### Quality and Clarity:

The paper is well-written and easy to follow.

### Typos:
-	Table 4 Column heading should be “Validation Unseen”. It is currently “Validation Seen”.


**Time Spent Reviewing:**

10

---

> ### Author Response · Authors · 2021-08-10
> **Author Responses to Reviewer JseM**
>
> **Q1: Table 4 Column heading should be “Validation Unseen”. It is currently “Validation Seen”.**
>
> A1: Thank you for pointing out the typo. We will correct it in the revision.

---

### Decision · Program_Chairs · 2021-09-27

**Decision:**

Accept (Poster)

**Comment:**

This work introduces a new resource for the English fold of RxR and shows how these annotations lead to significant gains in the original related domains/tasks.  The authors have provided substantial clarifications and new results during the discussion which are critical to understanding their work.  These need to be included in the final revision.  Note, in particular, many of the concerns raised by dQiv and the results for gPfu.  Some discussion of whether the resource can provide utility to the full RxR task (not just English) would also be appreciated.